# Reconfigurable image processing metasurfaces with phase-change materials

Michele Cotrufo [1,2,7] ✉, Shaban B. Sulejman [3,7], Lukas Wesemann [3,7], Md. Ataur Rahman[4], Madhu Bhaskaran[4,5], Ann Roberts [3] & Andrea Alù [1,6] ✉

Optical metasurfaces have enabled analog computing and image processing within sub-wavelength footprints, and with reduced power consumption and faster speeds. While various image processing metasurfaces have been demonstrated, most of the considered devices are static and lack reconfigurability. Yet, the ability to dynamically reconfigure processing operations is key for metasurfaces to be used within practical computing systems. Here, we demonstrate a passive edge-detection metasurface operating in the near-infrared regime whose response can be drastically modified by temperature variations smaller than 10 °C around a CMOS-compatible temperature of 65 °C. Such reconfigurability is achieved by leveraging the insulator-to-metal phase transition of a thin layer of vanadium dioxide, which strongly alters the metasurface nonlocal response. Importantly, this reconfigurability is accompanied by performance metrics—such as numerical aperture, efficiency, isotropy, and polarization-independence – close to optimal, and it is combined with a simple geometry compatible with large-scale manufacturing. Our work paves the way to a new generation of ultra-compact, tunable and passive devices for all-optical computation, with potential applications in augmented reality, remote sensing and bio-medical imaging.

Analog optical computing[1] has attracted renewed interest in recent years due to the exponentially growing demand for data processing[2], facilitating and accelerating computational tasks that would otherwise be performed electronically[3–5]. All-optical computational platforms offer the appealing possibility of manipulating data at the speed of light while avoiding analog-to-digital conversion[6], leading to large reductions in latency times and energy consumption. In this context, image processing is one of the most important computational tasks, underpinning many technologies such as augmented reality, self-driving vehicles, and LiDAR systems. These operations can be readily performed electronically, i.e., by digitalizing an image and manipulating it via a software. While easy to implement, digital approaches often suffer from several drawbacks, such as high latency times, energy consumption incompatible with stand-alone devices, and overall footprint and complexity, thus making analog alternatives particularly appealing and sought-after. The most commonly used approach to perform analog optical image processing is through Fourier filtering, using a so-called *4f* lens configuration[7]: an input image is Fourier-transformed by a first lens, and the different Fourier components are selectively filtered by a spatially varying amplitude and/or a phase mask placed in the Fourier plane. Finally, the output image is created via an inverse Fourier transform performed by a second lens. Edge detection, for example, can be performed with an opaque stop blocking low spatial frequencies and transmitting those above a

[1]Photonics Initiative, Advanced Science Research Center, City University of New York, New York, NY 10031, USA. [2]The Institute of Optics, University of Rochester, Rochester, NY 14627, USA. [3]ARC Centre of Excellence for Transformative Meta-Optical Systems, School of Physics, The University of Melbourne, Melbourne, VIC 3010, Australia. [4]Functional Materials and Microsystems Research Group and the Micro Nano Research Facility, RMIT University, Melbourne, VIC, Australia. [5]ARC Centre of Excellence for Transformative Meta-Optical Systems, RMIT University, Melbourne, VIC, Australia. [6]Physics Program, Graduate Center of the City University of New York, New York, NY 10016, USA. [7]These authors contributed equally: Michele Cotrufo, Shaban B. Sulejman, Lukas Wesemann. ✉e-mail: mcotrufo@optics.rochester.edu; aalu@gc.cuny.edu

certain threshold defined by the size of the opaque region. While easy to implement, the 4f approach is not suitable for integrated devices because it is inherently bulky and prone to alignment issues.

Recently, it has been shown that Fourier-based image processing can be implemented without the need of a bulky 4f system, by instead filtering the transverse momentum of an image directly in real space[8,9]. The implementation of this idea, sometimes referred to as an 'object' or 'image plane' approach[9], requires optical devices with a tailored angle-dependent response[10–24], which can be obtained with the aid of metasurfaces −ultra-thin films that are patterned on sub-wavelength scales. In particular, periodic nonlocal metasurfaces[10,11] can be engineered to perform momentum filtering within a very small footprint, as demonstrated by several theoretical[10–12] and experimental[17,19–24] studies. One of the most common image processing functionalities is the calculation of the spatial gradients of an input image $E_{in}(x,y)$, for instance by applying the Laplacian operation $E_{out}(x,y) = \nabla^2 E_{in}(x,y)$, which results in an enhancement of the edges of the input image as compared to areas of constant intensity (as sketched in Fig. 1a, left side). In Fourier space, the Laplacian operation is described by $E_{out}(k_x,k_y) = -(k_x^2 + k_y^2)E_{in}(k_x,k_y)$, that is, a high-pass filter that suppresses the small Fourier components corresponding to the flat features of the input image. Physically, this is achieved when the metasurface blocks (by reflection or absorption) any plane wave incident at small angles, while largely transmitting plane waves propagating at larger angles. This response can be achieved in metasurfaces supporting one or more dispersive optical modes[17,19,22–24], leading to

efficient edge-detection devices that do not require bias and operate in real space, without the need for a 4f system, thus drastically reducing the associated footprint.

While several studies[16–24] have demonstrated the possibility of performing image processing and edge detection in compact 4-less systems with the aid of metasurfaces or other approaches, most past work has considered 'static' devices, whose functionality is fixed and cannot be dynamically modified. Yet, such reconfigurability is necessary in order for these devices to replace digital computing in practical systems, particularly in the context of more complex optical computing tasks[25]. In order to achieve such reconfigurability, it is necessary to introduce a large, controllable and reversible change of the optical properties of the material forming the metasurface. For example, ref. 26 has demonstrated computational metasurfaces reconfigurable through mechanical strain, yet with limitations in terms of reproducibility and scalability. Several theoretical proposals[27–30] have suggested that large reconfigurability may be achieved by electrically gating a graphene layer placed above or inside a metasurface. Liquid crystals can also be used to achieve reconfigurable optical computation, although this approach typically requires thick devices used as spatially varying masks in 4f systems[31,32], thus limiting the miniaturization and integration opportunities. Material nonlinearities have also been proposed as a tool to achieve reconfigurability, either by having a high-intensity image impinging on a metasurface made of a saturable absorber[33], or by controlling the metasurface response via an external optical pump[34].

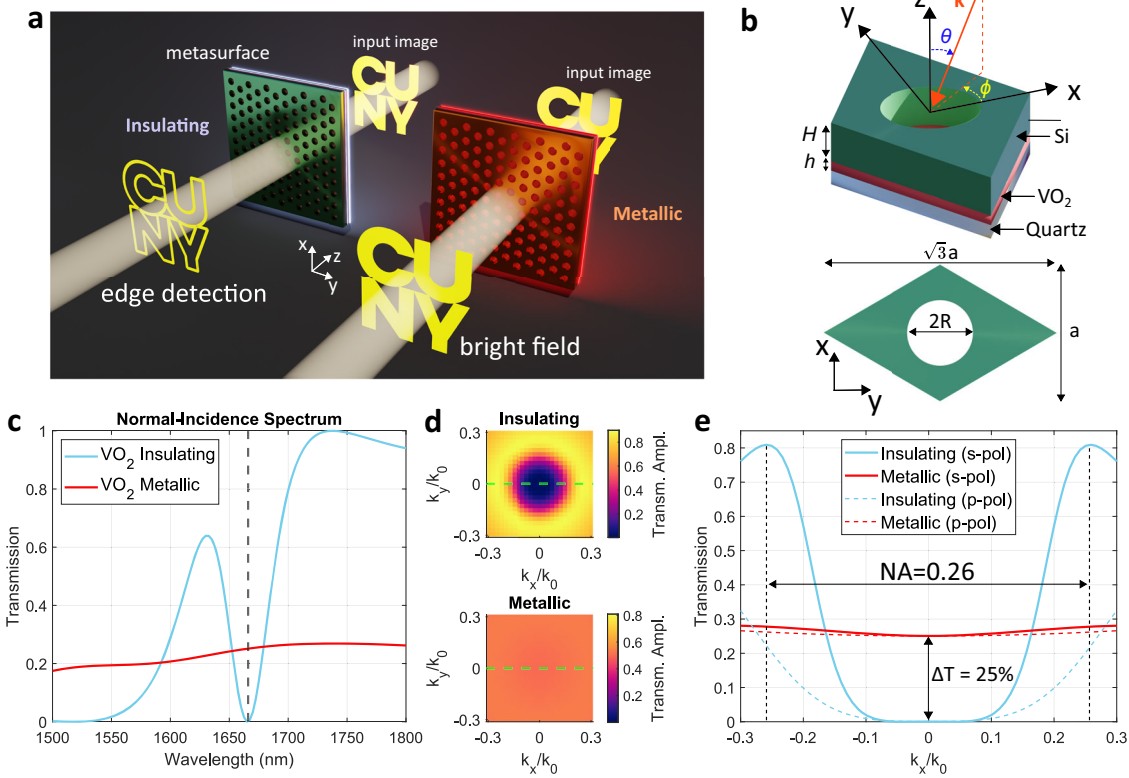

**Fig. 1 | General working principle and simulated optical response of the proposed metasurface. a** Schematic of the proposed working principle. The same metasurface performs either edge-detection (left) or conventional bright field imaging (right) depending on whether its temperature is lower or higher, respectively, than the transition temperature of a thin layer of vanadium dioxide (VO₂) embedded inside the metasurface. **b** 3D (top) and bird's eye (bottom) view of the unit cell of the device considered in this work. The metasurface is composed of a photonic crystal, made of a triangular lattice of holes etched onto a silicon slab, placed on a quartz substrate coated with a thin, unpatterned layer of VO₂. **c** Calculated normal-incidence transmission spectrum of the metasurface, assuming that the VO₂ is in the insulating (cyan curve) or metallic (red curve) phase. The normal-incidence spectra are independent of the excitation polarization due to the triangular rotational symmetry of the metasurface. **d** Angle-dependent transmission amplitude (absolute value) of the metasurface under s-polarized illumination, for the insulating (top) and metallic (bottom) phases of VO₂. **e** Transmission versus $k_x = k_0 \sin\theta$ for $k_y = 0$ for s-polarized (solid curves) and p-polarized (dashed curves) illumination, and for the insulating (cyan curves) or metallic (red curves) phases of VO₂.

In parallel, phase-change materials (PCMs) have emerged in recent years as an interesting platform to implement strong and controllable changes of the dielectric permittivity of nanophotonic structures. Materials such as vanadium dioxide ($VO_2$)[35], antimony trisulfide ($Sb_2S_3$), and germanium antimony telluride (GST), undergo an abrupt change in their crystalline structure as the temperature exceeds a certain threshold. Recent studies have investigated the integration of optical metasurfaces with $VO_2$[36,37] and GST[38,39] in order to dynamically modify the device response. In the field of imaging, ref. 40 proposed using germanium antimony selenide telluride (GSST) to realize tunable spectral filters for night vision applications. $VO_2$ is of particular interest in this context because its transition temperature is only a few tens of degrees above room temperature[35], which reduces the amount of energy required to change its permittivity, facilitating its implementation within more complex systems.

In this work, we merge the strong tunability provided by PCMs with the field of nonlocal dielectric metasurfaces, to realize an edge-detection 4f-less device whose image processing functionality can be dynamically reconfigured by a temperature change of few degrees. In particular, we experimentally demonstrate a single-layer metasurface, operating in the near-infrared, that performs high-efficiency and high-contrast edge detection at any temperature below the $VO_2$ transition temperature $T_0 \approx 65\,°C$, and whose image processing functionality can be drastically modified by varying the device temperature by less than 10 °C around $T_0$ (Fig. 1a). Such reconfigurability is achieved by altering the nonlocal response of the metasurface: in the insulating phase, the low loss of the $VO_2$ facilitates long-range interactions between multiple unit cells, giving rise to delocalized photonic modes whose dispersion and nonlocality are leveraged to accurately tailor the angle-dependent metasurface response. As the $VO_2$ transitions from the insulating to the metallic phase, the sudden increase in loss strongly inhibits the long-range interactions, reducing the degree of nonlocality and resulting in a device with an almost angle-independent transmission profile.

Besides the strong and controllable tunability, our proposed design simultaneously features remarkably large throughput efficiency, full isotropy, a relatively large NA ≈ 0.26 and an almost polarization-independent response. Our work presents a proof-of-principle implementation of reconfigurable image processing metasurfaces, which may be extended to devices capable of performing other temperature-controlled operations, including bandpass filtering, convolution, direction sensing, polarization imaging and quantitative phase microscopy. In particular, our device could also be used to perform reconfigurable phase imaging[20,23].

## Results
### Design and simulations

The metasurface design is based on a photonic crystal inspired by the geometry introduced in refs. 22,24 and illustrated in Fig. 1a, b. It consists of a silicon slab of height $H$ patterned with cylindrical holes of radius $R$ arranged in a triangular lattice with pitch $a$. The silicon slab rests on a uniform thin film of vanadium dioxide of thickness $h$, and the entire device is supported by a quartz substrate. The design was numerically optimized with a commercial electromagnetic solver (COMSOL Multiphysics) to obtain the desired reconfigurable response at wavelengths in the near-infrared range (1500 nm–1700 nm). The permittivities of the insulating and metallic phases of $VO_2$ were extracted from the literature[41], while the silicon and quartz were modeled with lossless and dispersionless refractive indices equal to 3.47 and 1.445, respectively. The design geometry was optimized such that, at a given operational wavelength, the angle-dependent transmission amplitude $t(\theta)$ of the metasurface supports a Laplacian profile [$t(\theta) \propto (\sin\theta)^2$] when the $VO_2$ is insulating, and an approximately flat profile [$t(\theta) \propto$ constant] when the $VO_2$ is metallic. By keeping the $VO_2$ layer very thin, we ensured that the absorption induced by this

material (particularly in the metallic phase) played a limited role in the overall device performance.

The normal incidence transmission spectra of an optimized device (with parameters $H = 320\,nm$, $R = 310\,nm$, $a = 960\,nm$, h = 35 nm) is shown in Fig. 1c for the insulating and metallic phases of $VO_2$. For insulating $VO_2$ (blue curve in Fig. 1c), the transmission spectrum features a high-contrast dip at wavelengths close to $\lambda = 1665\,nm$, which sets the operational wavelength. On the other hand, when the $VO_2$ is metallic, the transmission spectrum evolves into a significantly different lineshape (red curve in Fig. 1c), being almost wavelength-independent between 1500 nm and 1800 nm, and with an average transmission in the 20–30% range. This transmission level is primarily determined by the increased absorption and reflection induced by the metallic phase of $VO_2$. We note that, owing to the C6 rotational symmetry of our design, the normal-incidence transmission spectra in Fig. 1c are independent of the polarization state of the illumination. However, polarization dependence is expected to arise as the polar angle $\theta$ is increased. To verify that this design can perform reconfigurable edge detection, we numerically calculated the angle-dependent transmission of the metasurface for the two phases of $VO_2$ (Fig. 1d, e). In Fig. 1d we show the absolute value of the s-polarized angle-dependent transmission amplitude at a fixed wavelength ($\lambda = 1665\,nm$) and for the two phases of $VO_2$ (the corresponding phases of these transmission amplitudes are shown in the Supplementary Information, Section S2). When the $VO_2$ is in the insulating phase (top of Fig. 1d), the metasurface supports an isotropic Laplacian-like response within a numerical aperture NA ≈ 0.26. The absolute value of the transmission amplitude evolves from almost zero at normal incidence to values as high as 0.9 (corresponding to transmission of about 81%) for polar angles $\theta = 15°$. Such a response indicates that, when the $VO_2$ is insulating, the metasurface performs high-efficiency edge detection on an input image. On the other hand, when the $VO_2$ is in the metallic phase (bottom of Fig. 1d), the absolute value of the transmission amplitude is pinned to a value of approximately 0.5 (corresponding to a transmission of about 25%) for any angle of incidence within the numerical aperture. Such an angle-independent response indicates that the metasurface will not perform Fourier filtering, faithfully reproducing the input image (albeit with some intensity attenuation). The p-polarized transfer functions (shown in the Supplementary Information, Section S2) display a similar behavior, although with a reduced value of transmission at large angles when the $VO_2$ is insulating. The reconfigurability of the metasurface transfer function is further demonstrated in Fig. 1e, which shows the calculated metasurface transmission as a function of $k_x = k_0 \sin\theta$ and for a fixed $k_y = 0$, for both s- and p-polarized illuminations at the operational wavelength. Here, $k_0 = 2\pi/\lambda$ denotes the wavenumber in free space and ($k_x, k_y, k_z$) are the Cartesian components of the wave-vector **k**.

As shown in Fig. 1e, for both s and p impinging polarizations the metasurface transfer function features a flat profile when the $VO_2$ is in the metallic phase. When the $VO_2$ is in the insulating phase, both s- and p-polarized transfer functions feature the desired quasi-parabolic angle-dependent transmission, albeit with different magnitudes: the s-polarized transfer function reaches values larger than 0.8 at $k_x/k_0 = 0.25$, while the p-polarized transfer function is about 0.2 at the same excitation angle. As discussed in ref. 24, such polarization asymmetries do not introduce any practical issue in the quality and uniformity of the edge-detected image if the input image is unpolarized. Instead, when the input image is linearly polarized, large asymmetries between s- and p-polarized transfer functions could lead to a scenario where different edges are enhanced with different efficiencies depending on their orientation[24]. In the experiments discussed here, we focus on unpolarized input images. To complement the experimental results, in the Supplementary Information (Section S4) we show with numerical calculations that, even when the input image is

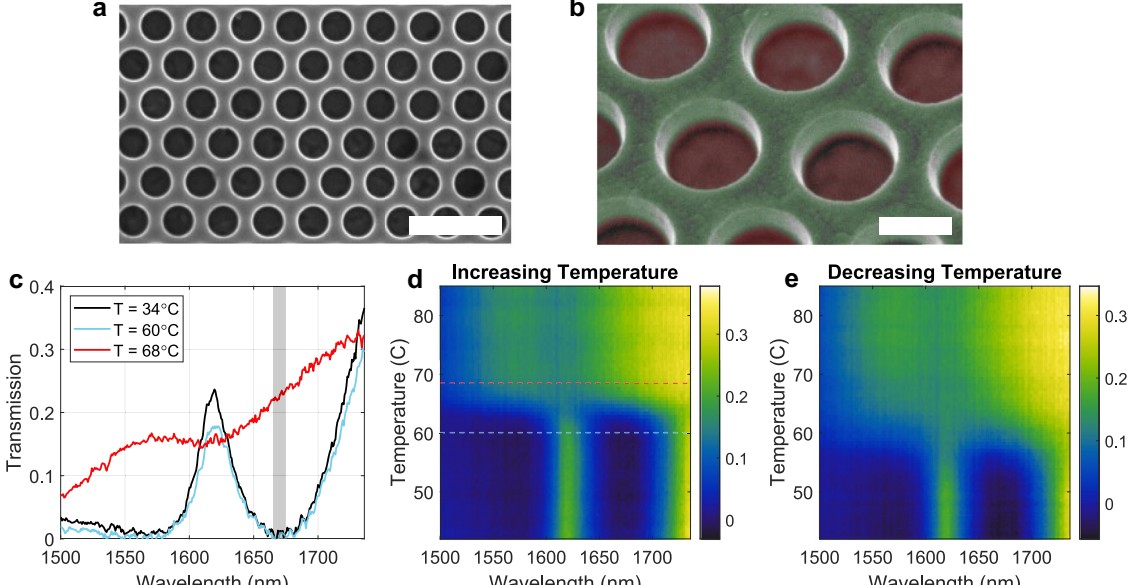

**Fig. 2 | Experimentally measured optical response of the fabricated metasurface.** Scanning electron microscope images of the fabricated metasurface. Scale bars = 2 μm in (**a**) and 500 nm in (**b**). The false-colored image in (**b**) shows the silicon slab (green) and the underlying VO$_2$ film (red). **c** Transmission spectrum of the device selected for further experiments, at near-room temperature (T = 34 °C, black curve), at T = 60 °C (cyan curve) and at T = 68 °C (red curve). Evolution of the normal-incidence transmission spectrum of the metasurface in (**c**), as the temperature is slowly increased (**d**) and decreased (**e**).

polarized, the polarization asymmetry of our device (Fig. 1e) is not large enough to introduce any visible inhomogeneity or anisotropy.

## Experimental results

We fabricated a metasurface with parameters close to the numerically optimized design. The fabrication was performed with standard lithographic and etching techniques, as described in the Methods. Figure 2a, b show SEM images of a fabricated sample with hole radius $R \approx 350$ nm, VO$_2$ thickness $h \approx 35$ nm, silicon thickness $H \approx 360$ nm, and lattice constant $a = 960$ nm, which we selected for the optical characterization and experiments. In the false-colored tilted image in Fig. 2b, the silicon photonic crystal is colored in green, while the underneath VO$_2$ layer is colored in red. We used a custom-built optical setup to characterize the optical response of the metasurface as a function of wavelength, incident angle, and temperature (see Methods and Supplementary Information, Section S1, for additional details). The device temperature was controlled by placing it in contact with a thin ceramic heater, and the temperature of the metasurface was monitored by both a thermal camera and a temperature detector attached on the silicon side of the device.

The black line in Fig. 2c shows the measured normal-incidence transmission spectrum of the metasurface at temperatures close to room temperature (T = 34 °C). The transmission spectrum is characterized by a minimum between 1662 nm and 1675 nm (gray shaded area), where the transmission drops below 2%, in good agreement with simulations. Figure 2d shows the evolution of the normal-incidence transmission spectrum as the temperature of the metasurface is slowly increased. For temperatures ranging from room temperature to approximately 60 °C, the transmission spectrum remains almost unchanged, featuring the same minimum in the 1662 nm–1675 nm region. To further confirm this, the transmission spectrum at T = 60 °C (corresponding to the dashed cyan line in Fig. 2d) is plotted in Fig. 2c (cyan line). Very small changes of the transmission spectrum for temperatures ranging from room-temperature to 60 °C are due to a weak and continuous variation of the refractive index of silicon due to thermo-optic effects.

As the temperature of the metasurface is increased above 60 °C, the VO$_2$ undergoes an insulator-to-metal transition, causing the transmission spectrum to rapidly morph into a different line shape. In this state, the transmission minimum disappears, and the wavelength dependence of the device becomes almost flat, as also shown by the horizontal cut at the temperature T = 68 °C in Fig. 2c (red curve, corresponding to the dashed red line in Fig. 2d). After this transition, for any temperature above 68 °C the transmission spectrum does not vary significantly. The temperature dependence of the transmission spectrum shown in Fig. 2c, d agrees well with the simulated response in Fig. 1c–e. In our measurements, the optical spectrum evolves smoothly within a narrow range of temperatures $\Delta T \leq 8$ °C instead of abruptly changing at a certain threshold temperature, as expected for an ideal phase change. This behavior can be attributed to inhomogeneities in the VO$_2$ film that introduce local variations of the threshold temperature[42]. The results confirm that the VO$_2$ phase transition lies within the temperature range 60–68 °C, in agreement with commonly reported value[35]. To demonstrate the process reversibility, the heater was turned off at the end of the measurement run of Fig. 2d, and the transmission spectrum was continuously recorded as the metasurface cooled to room temperature. The results (Fig. 2e) show that the original transmission spectrum is recovered when the metasurface cools to temperatures below ~55 °C, confirming that the change of the metasurface spectrum induced by the phase transition is fully reversible. A small hysteresis behavior is also observable when comparing Fig. 2d, e: the temperature at which the VO$_2$ undergoes phase transition is slightly different for the warming up and cooling down experiments. Similar hysteresis effects in VO$_2$ have been reported in previous work[42–44], and they can be attributed to the formation and interaction between clusters of domains in the metallic phase of the material[43].

Next, we verify that, besides the normal-incidence response, the phase transition of VO$_2$ can also largely reconfigure the nonlocal response of the metasurface—manifested in the angle-dependent transmission amplitude—which is crucial to modify the metasurface image processing functionality. To this aim, we performed temperature- and angle-dependent s-polarized transmission measurements for

a fixed wavelength λ = 1672 nm and azimuthal angle $\phi = 0°$, with results shown in Fig. 3a. For all temperatures below T = 60 °C, the angle-dependent transmission profile displays the desired Laplacian-like behavior, evolving from a transmission of about 2% at normal incidence to a transmission larger than 50% for $\theta \approx 16°$. These transmission levels agree with our simulations (Fig. 1), and they are comparable with the ones obtained in similar non-reconfigurable devices[22,24]. This confirms that the absorption due to the optical losses within the insulating VO$_2$ is negligible in this wavelength range. As shown by the three cyan lines in Fig. 3b, the quadratic-like angle-dependent transmission profile remains almost unchanged for temperatures T < 60 °C. Minor variations in the angle-dependent transmission profiles for T < 60 °C (e.g., the small decrease of the transmission peak at $\theta \approx 16°$ for T = 59 °C) can

be attributed to the weak and smooth variation of the silicon permittivity due to thermo-optic effects. For temperatures T > 60 °C, the angle-dependent transmission undergoes a quick and sudden change, evolving to an almost flat profile which is maintained for any temperature T > 68 °C (see also the three red lines in Fig. 3b). Overall, the experimental data in Figs. 2 and 3 confirm that the VO$_2$ phase transition can be leveraged to induce a drastic change in the metasurface nonlocal response. In particular, the high-pass-filter response, which is the mathematical basis for edge detection, can be turned on and off by small ($\Delta T \leq 8° C$) changes in temperature. This discrete and sudden change in nonlocality and angle-dependent response is very different from what can be obtained with typical thermo-optic effects[45], whereby the refractive index changes in a continuous fashion as a function of the temperature, and thus temperature changes of several tens of degrees are necessary to obtain significant spectral changes.

After having verified the temperature-induced reconfigurability of the metasurface nonlocality, we now demonstrate that this platform can be used to realize an image processing device with edge-detection functionality that can be rapidly switched on and off with a temperature change of a few degrees. The imaging experiments were performed with the setup shown in Fig. 4a, and further detailed in the Supplementary Information (Section S1). Test input images were created by illuminating an amplitude mask with collimated and unpolarized light with a central wavelength of 1670 nm and a linewidth of about 5 nm. The mask was created by etching a desired shape onto a 200 nm thick layer of chromium deposited on a glass substrate. The image scattered by the target was collected by a NIR objective (Mitutoyo, 50X, NA = 0.42) and relayed onto a near-infrared camera. The metasurface was mounted on the heating stage used for the measurements in Figs. 2 and 3 and placed between the mask and the objective. The heating stage was placed on a flip mount, allowing us to

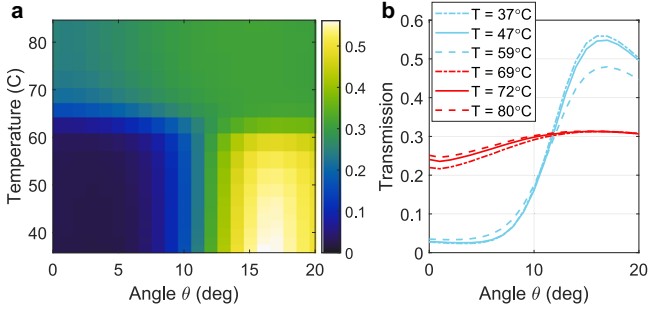

**Fig. 3 | Thermal control of the metasurface filtering response. a** Measured temperature- and angle-dependent transmission of the device in Fig. 2 for s-polarized illumination at a wavelength of λ = 1672 nm. **b** Measured transmission versus angle for selected values of the temperature (extracted from **a**), as denoted in the panel legend.

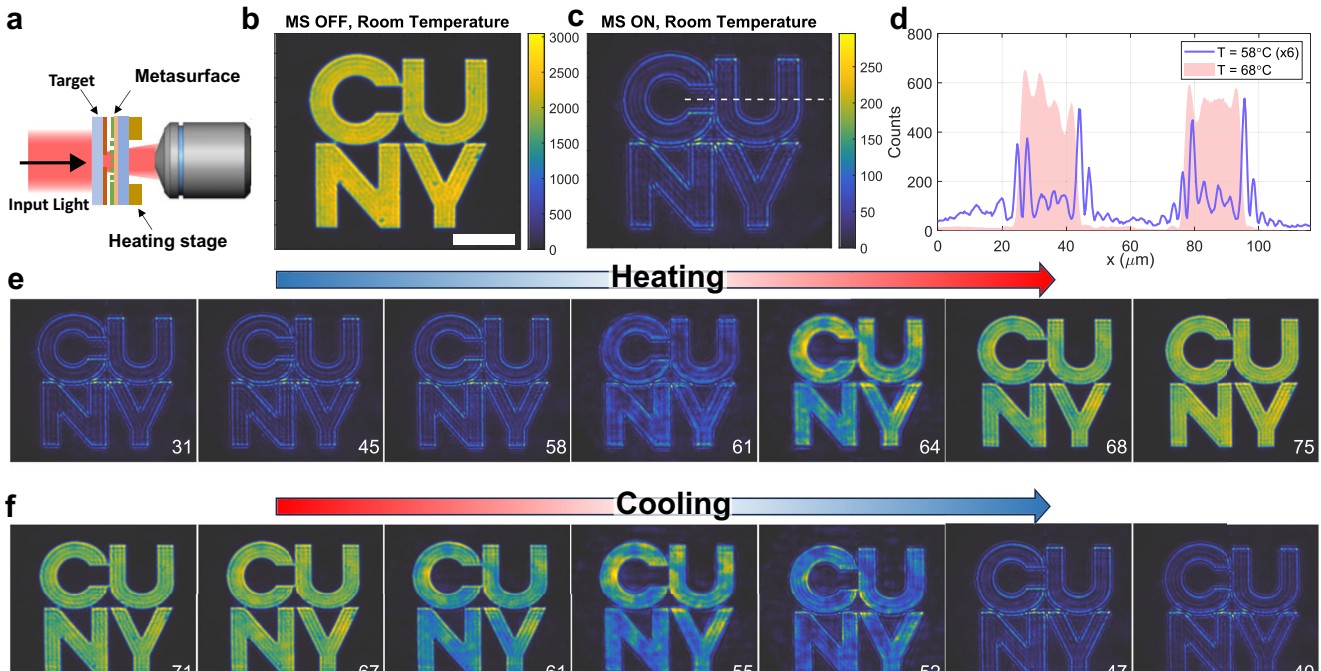

**Fig. 4 | Reconfigurable edge-detection experiment. a** Schematic of the experimental setup. **b** Unfiltered input image (corresponding to the logo of our institution) obtained when the metasurface is removed from the setup in panel a (scalebar = 50 μm). **c** Edge-detected image obtained with the metasurface present and at room temperature. **d** Horizontal slices of the output intensity taken at the location marked by the white dashed line in (**c**), for a metasurface temperature slightly below

(T = 58 °C, blue curve) and slightly above (T = 68 °C, red shaded areas, extracted from the corresponding image in **e**) the transition temperature of VO$_2$. The blue curve has been rescaled by a factor of 6x to aid comparison. **e** Filtered images obtained for different temperatures of the metasurface as the device is heated. The temperature of each measurement is reported in the bottom right corner of each image. **f** Same as in panel e but for the case in which the device is cooled.

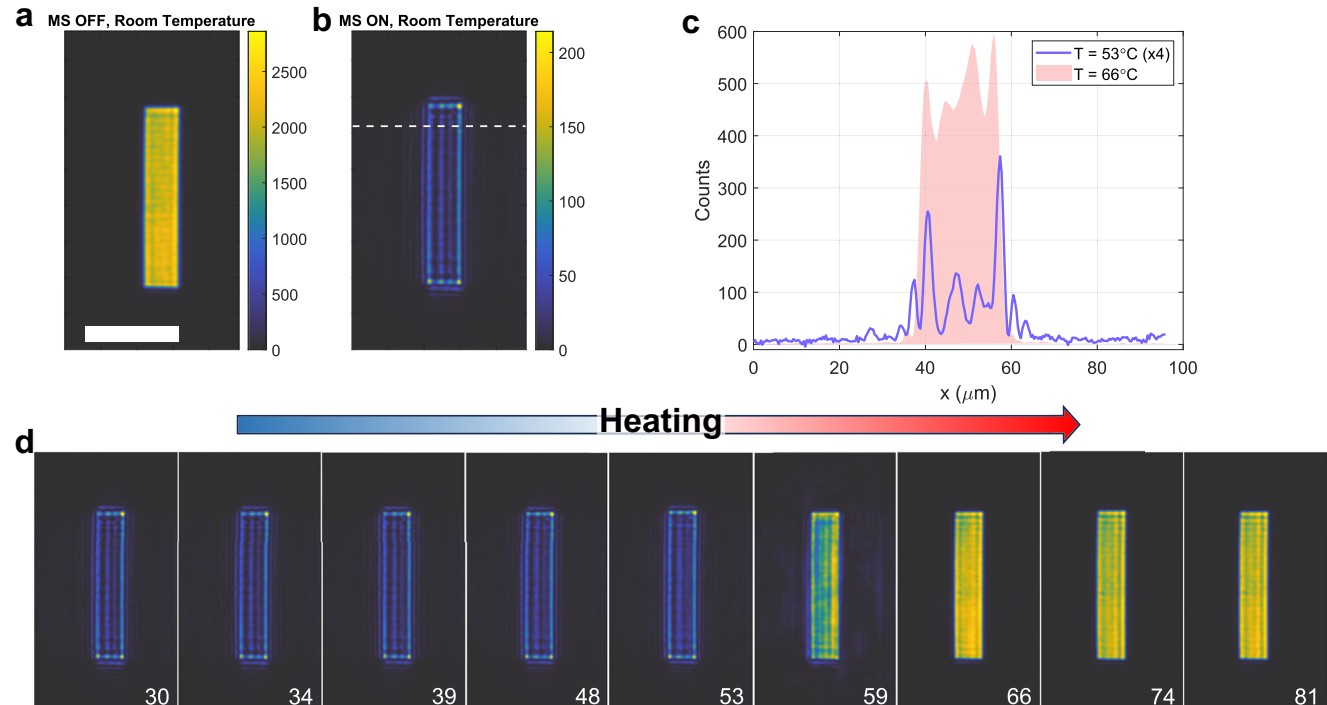

**Fig. 5 | Reconfigurable edge-detection experiment with a rectangular input image. a** Unfiltered input image obtained in the absence of the metasurface (scale bar = 50 μm). **b** Edge-enhanced image obtained with the metasurface at room temperature. Horizontal slices of the output intensity taken at the location marked by the white dashed line in (**c**), at a temperature slightly below (T = 53 °C, blue curve) and slightly above (T = 66 °C, red shaded area) the transition temperature of vanadium dioxide. The blue curve has been rescaled by a factor of 4x to comparison. **d** Filtered images versus temperature as the metasurface is heated. The temperature of each measurement is shown in the bottom right corner of each image.

relay onto the camera either the unfiltered input image (when the metasurface was removed) or the input image filtered by the metasurface.

The imaging results are summarized in Figs. 4 and 5. In Fig. 4, the target was the logo of one of our institutions (CUNY) with a transverse size of approximately 150 μm. Figure 4b shows the output image recorded by the camera when the metasurface is removed from the setup in Fig. 4a. In this scenario, the full unfiltered image is recorded. Next, we inserted the metasurface between the mask and the objective, and we recorded the corresponding image with the device at room temperature (Fig. 4c). As expected, in this configuration the edges of the input image are strongly enhanced with respect to the regions where the intensity is homogenous. An important figure of merit for passive computational metasurfaces is the throughput efficiency, that is, how the intensity of the output image compares to the intensity of the input image. Following ref. 22, we define the peak efficiency as the ratio between the maxima of the input and output intensities $\eta_{peak} \equiv \max(I_{out})/\max(I_{in})$. In order to readily estimate this metric, the color plots in Fig. 4b, c have been normalized by dividing the counts recorded in each pixel by the power incident on the target and by the integration time of the camera. Thanks to this normalization, the pixel intensities in both input and output images are proportional to the optical fluence, and the efficiency $\eta_{peak}$ can be readily estimated by comparing the upper limits of the colorbars in Fig. 4b, c. In particular, the intensity efficiency of our metasurface reaches values close to 10%. These efficiencies are similar to those measured in similar passive devices without the additional tunability provided by the PCM[22,24]. Moreover, this efficiency level is close to the maximum possible efficiency attainable by an ideal filter performing the Laplacian operation with the same NA[22], ultimately limited by the passivity of these devices.

After having verified that our device can perform high-quality and isotropic edge detection at room temperature (Fig. 4c), we investigated how the output filtered image depends on the metasurface temperature. To this aim, the image was continuously acquired as the temperature of the metasurface was increased and then subsequently decreased. In Fig. 4e, f we show the output image recorded at seven different temperatures, for increasing (Fig. 4e) and decreasing (Fig. 4f) temperatures (in the Supplementary Information, Section S3, we display additional images acquired at other temperatures). In each subpanel, the corresponding temperature (in degrees Celsius) is shown in the bottom left corner. As clearly shown by Fig. 4e, the output image remains unaltered for any temperature below 61 °C, displaying the same high-contrast edge enhancement achieved at room temperature (Fig. 4c). For temperatures above $T = 61°C$, the output image quickly evolves and, for temperatures above $T = 68°C$, the edge enhancement has completely disappeared, and the output image now closely resembles the input (Fig. 4b). The changes in the output image between the two states can be further visualized in Fig. 4d, which shows horizontal slices of the output intensity maps (taken along the horizontal dashed line in Fig. 4c) for the temperatures $T = 58°C$ (blue solid curve) and $T = 68°C$ (red shaded area). As the device is cooled down (Fig. 4f), the edge-detecting functionality is fully restored as soon as the temperature drops below 52°. The hysteresis behavior – that is, the difference in transition temperature between the heating (Fig. 4e) and cooling (Fig. 4f) experiments—is similar to that observed in Fig. 2d, e. Other than the presence of hysteresis, the reconfigurability of the edge-detecting functionality is fully reversible, as confirmed by Fig. 4f. In particular, the peak intensities of the last two panels of Fig. 4f are the same as the peak intensities of the first three panels of Fig. 4e (see Supplementary Information, Section S3, for full data), confirming that no degradation has occurred in the metasurface. For temperatures within the transition region some image distortion is noticeable in the output images (e.g., 64 °C in Fig. 4e and 55 °C in Fig. 4e). This effect can be attributed to the aforementioned inhomogeneities in the VO₂ film that introduce local variations of the threshold temperature[42]. As a result, different portions of the metasurface will

experience the insulating-to-metal transition at slightly different temperatures, resulting in an undesired image distortion. These effects can be mitigated by working with higher-quality layers of phase-change materials.

As shown in Fig. 4d, the edge-detection contrast—defined as the ratio between the intensity of the detected edges and the intensity of the surrounding background areas—can be quite large, reaching values of about 10. The edge-detection contrast is directly related to how well the metasurface can suppress the low-spatial-frequency components of the input image, which in turn is dictated by the normal-incidence transmission level. In our device, we achieved normal-incidence transmissions of about 2%–3% (see Fig. 3b), mainly limited by the small absorption of the insulating $VO_2$ and by fabrication imperfections. Improvements of these factors would strongly reduce the residual background in the output images. Figure 4d also reveals the presence of additional weaker intensity fluctuations between each pair of edges, both in the input and output images. Such weak intensity modulations are due to the fact that very-high spatial-frequency components, induced by the small size of the input image and the presence of sharp edges, are clipped by the finite objective NA (0.42) and metasurface NA (0.26). Nonetheless, the presence of these additional weaker oscillations does not limit the capability of performing edge detection, since the peaks corresponding to the edges are always much more intense.

To further verify the reconfigurable image processing capabilities of our device, the imaging experiment was repeated with a different input image, given by a rectangle of dimensions $20\,\mu m \times 100\,\mu m$ (Fig. 5a). The image filtered by the metasurface at room temperature (Fig. 5b) shows high-quality edge enhancement, which was maintained unaltered for all temperatures up to $T \approx 60\,°C$ (Fig. 5d). Similar to the results observed in Fig. 4, the output image undergoes a sudden change as the temperature increases above $T = 60\,°C$, morphing into the unfiltered attenuated image for temperatures above $T = 66\,°C$. A comparison between the intensity profiles for $T = 53\,°C$ and $T = 66\,°C$ is shown in Fig. 5c.

We note that, while in our experiments we focused on edge detection, the *same* device can also be used to switch between phase imaging[20,23] and bright field imaging, which is of particular relevance for biological samples. This is possible due to the fact that, in these analog devices, the Laplacian operation is performed at the level of the field amplitude, i.e., $E_{out}(x,y) = \nabla^2 E_{in}(x,y)$. Thus, if the input field has a constant amplitude but a spatially varying phase, $E_{in}(x,y) = Ae^{i\phi(x,y)}$, the output field will be proportional to the Laplacian of $\phi(x,y)$, i.e., $E_{out}(x,y) \propto \nabla^2 \phi(x,y)$. Thus, the output intensity will carry information about the phase gradients and discontinuities of the input image.

An important figure of merit for reconfigurable metasurfaces is the switching time, i.e., how long it takes for the metasurface to switch between two states once an external stimulus is applied. In our PCM-based device, the switching time is due to the sum of two components, $t_{switch} = t_{heat} + t_{PCM}$. The time $t_{PCM}$ is the time that the phase-change material takes to transition from insulating to metal once the transition temperature is achieved. Several works[46–48] have found that $t_{PCM}$ can be shorter than 1 ps, with values of the order of few tens of femtoseconds found in ref. 48. These timescales would potentially allow PCM-based metasurfaces to perform ultrafast signal processing. However, in practical devices the overall transition time is actually dictated by $t_{heat}$, which is the time required to change the sample temperature from a temperature $T_{cold}$ to a temperature $T_{hot}$ (or vice versa), where $T_{cold}$ and $T_{hot}$ are just below and above the transition range, respectively. In our experiment, $T_{cold} \approx 60\,°C$ and $T_{hot} \approx 68\,°C$, and thus $t_{heat}$ is the time required to change the metasurface temperature by approximately 8 degrees Celsius. The speed at which this temperature change occurs depends on how the sample is heated. In our proof-of principle experiment, we heat the sample with an external heater, which is contact with the backside of the sample. Thus, the metasurface heating

timescales are dictated by the heat conduction through the thick (1 mm) glass substrate, and therefore they are expected to be fairly slow, typically on the ~1 s scale (see also Supplementary Video 1). We note, however, that this is not an intrinsic limitation of this approach. More efficient heating mechanisms can be devised, for example by integrating metallic heating elements directly on top of the metasurface layer. In ref. 49 this approach was used to vary the temperature of a silicon metasurface by more than 50°C within a time <1 ms by using short 5 V voltage pulses. Moreover, phase transitions can also be induced electrically and without the need of any significant Joule heating effect[50], leading to switching timescales of the order of hundreds of microseconds. Additionally, all-optical heating, where a high-power laser alters the metasurface temperature, can be used to achieve even faster heating timescales. For example, in ref. 51 it was shown that a picosecond laser with an average power of ~100 mW can heat up a silicon metasurface by ~40 K within few tens of microseconds.

## Discussion

In this work we have demonstrated that phase-change materials can be used to drastically control the nonlocality and the angle-dependent transmission profile of metasurfaces, leading to sub-wavelength passive edge-detection devices whose image processing operation can be efficiently controlled by temperature variations smaller than 10 °C around a CMOS-compatible temperature $T_0 \approx 65\,°C$. This reconfigurability is achieved by leveraging the insulating-to-metallic phase transition of a layer of $VO_2$ arising at temperatures close to $T_0$. Our design principle, based on adding a thin layer of $VO_2$ within a thicker metasurface, enables a large change of the optical properties of the metasurface while simultaneously minimizing absorption losses when the $VO_2$ is in the insulating phase.

In particular, we have designed and experimentally demonstrated a metasurface that, for any temperature $T < 60\,°C$, features the same Laplacian-like angle-dependent transfer function within a numerical aperture of 0.26, well suited to perform high-resolution edge detection. As the temperature increases and the $VO_2$ undergoes the phase change, the metasurface transfer function rapidly evolves into an almost angle-independent profile, with average transmission around 25%, which is maintained for any temperature $T > 68\,°C$. In this state, the metasurface does not perform any operation on the input image. Moreover, the proposed reconfigurable metasurface performs high-NA, high-efficiency, isotropic, and polarization-independent edge detection with metrics close to optimal, and close to the ones demonstrated in recent works with similar devices that lack reconfigurability[22,24]. The relatively large NA ≈ 0.26 ensures that our metasurface can perform edge-detection with high spatial resolution, which can be estimated via $R \geq \frac{\lambda}{2NA} \approx 3.2\,\mu m$. Indeed, in our processed images (see, e.g., Fig. 4c) we are able to fully resolve edges that are ~10 μm apart from each other.

Our approach and design make this reconfigurable image processing metasurface amenable to mass manufacturing. In particular, differently from previous proposals, our approach does not require any mechanical and/or moving parts, electrical biases, or high-power optical excitations. Moreover, the transition temperature can be further reduced by doping $VO_2$ with molybdenum or tungsten to facilitate optical-induced heating process[52,53].

Importantly, the computational metasurface demonstrated in this work does not require the use of a $4f$ lens system, because the desired mathematical operation is implemented directly in real space, by filtering different plane waves with respect to their angle of propagation. This is in stark contrast to approaches where either liquid crystals[31,32] or metasurfaces[54–57] are used as spatially varying masks in a $4f$ system, which preclude any meaningful miniaturization.

Further improvement of the proposed design could lead to the implementation of more complex responses whereby, for example,

the metasurface performs two different mathematical operations of choice in the two states. To obtain a more advanced functionality at high temperatures (as opposed to the bright-field imaging demonstrated here), the metasurface must support a nontrivial angle-dependent transfer function also when the PCM has transitioned to the high-temperature state. In the current experiment, the angle-dependent transfer function at high temperature (Figs. 1e, 3b) is limited by the large loss induced by the metallic phase of the $VO_2$. Other PCMs with different types of phase transitions could be used instead of $VO_2$, in order to minimize the impact of loss. For example, Antimony Trisulfide (Sb2S3) features an amorphous-to-crystalline phase transition characterized by a large change of the real part of the refractive index ($\Delta n = n_{crystalline}/n_{amorphous} > 1.2$ at $\lambda \approx 1550$ nm), while the imaginary part of the refractive index remains $\kappa < 10^{-5}$ in both states[58]. Additionally, the approach demonstrated here can be extended to non-volatile PCMs[59], which would permit maintaining the desired functionality without actively heating the metasurface. Finally, in this device the temperature was controlled via external heater elements, which set hard bounds on the rate at which the temperature increases, and thus on the switching speed. However, the same working principle and metasurface design can be readily extended to devices where the metasurface temperature is controlled either by local heater elements integrated on the same chip[49,59], electrical effects[50], or via optically induced heating with an external pump laser[51]. The latter scenario may open interesting avenues for all-optically reconfigurable nonlinear analog computation. We expect these results to pave the way for the use of reconfigurable 4f-less image processing metasurfaces for applications such as augmented reality, satellite systems and environmental monitoring, as well as materials research.

## Methods

### Sample fabrication

The samples were fabricated with a standard lithographic process. First, $VO_2$ thin films were deposited onto plasma-cleaned fused silica substrates by using pulsed direct-current magnetron sputtering (Kurt J. Lesker PVD 75), as described in more detail in refs. 35,44. The sputtering was performed with base pressure of $4.0 \times 10^{-7}$ Torr, sputtering pressure of $2.8 \times 10^{-3}$ Torr, power of 200 W, and with 30% oxygen partial pressure in an argon environment. The as-deposited samples were annealed at 560 °C and 250 mTorr pressure for 1.5 h to obtain crystallinity. The $VO_2$-coated substrates were further cleaned by placing them in an acetone bath inside an ultrasonic cleaner, and in an oxygen-based cleaning plasma (PVA Tepla IoN 40). After cleaning, a layer of 360 nm of amorphous silicon was deposited via a plasma-enhanced chemical vapor deposition process. A layer of e-beam resist (ZEP 520-A) was then spin-coated on top of the samples, followed by a layer of an anti-charging polymer (DisCharge, DisChem). The desired photonic crystal pattern was then written with an electron beam tool (Elionix 50 keV). After ZEP development, the pattern was transferred to the underlying silicon layer via dry etching in an ICP machine (Oxford PlasmaPro System 100). The resist mask was finally removed with a solvent (Remover PG).

### Numerical simulations

The numerical simulations were performed using a finite-element-method commercial electromagnetic solver (COMSOL Multiphysics 6.1). Floquet boundary conditions were applied to the sides of the rhomboidal unit cell (shown in Fig. 1b) to model an infinite triangular lattice. The quartz substrate was assumed to be a homogeneous, lossless material with a refractive index of 1.445. The silicon material was assumed to be homogeneous and lossless, with a refractive index of 3.47 at the wavelength of 1665 nm. The $VO_2$ was modeled using ellipsometry data for both of the insulating and metallic states[41], and it was assumed to be lossless in the insulating state. Port boundary conditions were applied on the top and bottom of the unit cell to

launch and absorb plane waves, respectively. The complex angle-dependent transmission amplitude was obtained by computing the complex coefficient $S_{21}$ of the scattering matrix of the two-port system.

### Optical characterization

The measurements shown in Figs. 2–5 of the main paper were performed with a custom-built setup described briefly here below and in more detail in the Supplementary Information (Section S1 and Fig. S1).

For all measurements, the sample was attached to a thin ceramic heater (Thorlabs, HT10KR1). The temperature of the heater was increased by progressively increasing a current fed to it, and the temperature of the sample was monitored by both a platinum-resistance temperature detector (Thorlabs, TH100PT) attached on the silicon side of the metasurface and a thermal camera. The heater was placed on two independent rotation stages to control the polar angle $\theta$ and the azimuthal angle $\phi$. For the normal-incidence measurements shown in Fig. 2a, the output of an unpolarized and collimated broadband lamp was weakly focused on the metasurface, and the transmitted beam was collected by a near-infrared spectrometer. The measurements in Fig. 2d were obtained by slowly increasing the current fed to the heater, and by continuously recording the transmission spectra and the device temperature. After the temperature reached a value of approximately 90 °C, the current fed to the heater was turned off, and the transmission spectrum of the metasurface was continuously recorded as the sample cooled down to room temperature. For the angle-dependent measurements shown in Fig. 3, a broadband supercontinuum laser (NKT, Fianium FIU-15) filtered via a tunable narrowband filter (Photon, LLTF Contrast) was used as source. The laser was weakly focused on the metasurface, and the transmitted signal was collected and re-collimated on the other side of the sample by an identical lens. Two identical germanium power meters were used to measure the transmission level through the metasurface. A linear polarizer placed before the beam-splitter was used to polarize the incoming beam along either x or y, which correspond, respectively, to p- and s-polarization for any value of $\theta$ and $\phi$. A second linear polarizer was used to select the output polarization. The transmission amplitudes shown in Fig. 3a were then obtained with an automatized procedure whereby the temperature was slowly increased in small steps and, after achieving a thermal steady state, the angle $\theta$ was swept.

The imaging experiments shown in Figs. 4 and 5 of the main paper were performed with the setup schematized in Fig. 4a and shown in more details in Figure. S1. The illumination was provided by the same filtered supercontinuum source used in the setup described in the previous paragraph.

### Reporting summary

Further information on research design is available in the Nature Portfolio Reporting Summary linked to this article.

## Data availability

Data underlying the results presented in this paper may be obtained from the authors upon request.

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

## Acknowledgements

This work was supported by the Air Force Office of Scientific Research MURI program and the Simons Foundation. This work was also supported by the Australian Government through the Australian Research Council Centre of Excellence grant (CE200100010). S.B.S. acknowledges the support of the Ernst & Grace Matthaei Scholarship and the Australian Government Research Training Program Scholarship.

## Author contributions

M.C., S.B.S., L.W., A.R. and A.A. conceived the original idea and the experiment. S.B.S., M.C. and L.W. performed the design optimization. M.A.R. and M.B. prepared the VO$_2$-coated quartz substrates. M.C. fabricated the metasurfaces and built the optical setup. M.C. and S.B.S. performed all optical measurements. M.C., S.B.S. and L.W. analyzed the data, prepared the figures, and wrote the first draft of the manuscript. All authors contributed to finalizing the manuscript. A.A. and A.R. supervised the project.

## Competing interests

The authors declare no competing interests.
