## [Peer Review File · Nature Communications]

REVIEWER COMMENTS

Reviewer #1 (Remarks to the Author):

Optical metasurfaces, as a class of optical metamaterials with a reduced dimensionality, have attracted much attentions due to their fascinating abilities of controlling light. Recently, metasurfaces have started making important progress in of analog computing and image processing. Various image processing and optical computing functionalities have been recently demonstrated, however, most of the considered devices are static and lack reconfigurability. In the present manuscript, the authors propose and demonstrate a reconfigurable image processing metasurface phase-change material. In the proposed scheme, the image processing response can be drastically modified by temperature variations around a CMOS-compatible temperature. In addition, this reconfigurability is accompanied by performance metrics close to optimal, and it is combined with a simple geometry compatible with large-scale manufacturing. There are several issues in the paper need to be elucidated further.

In the section of Introduction, the authors write: Liquid crystals can also be used to achieve reconfigurable optical computation, although this approach typically requires thick devices used as spatially varying masks in 4f systems, thus limiting the miniaturization and integration opportunities. It should be noted that the fast switching with several millisecond switching time can be realized by liquid crystals. However, it is not clear how fast the switching time of the phase-change material is. If the switching time is not very fast, then what is the motivation of why a phase-change material is a good candidate in reconfigurable image?

The authors described the fabrication of the imaging target, which is created by etching the desired shape on a 200 nm thick chromium layer deposited on a glass substrate. Therefore, it is known that the amplitude target has been applied in the reconfigurable imaging experiment. In the abstract, the authors write: Our work paves the way to a new generation of ultra-compact, tunable, passive devices for all-optical computation, with potential applications in augmented reality, remote sensing, and bio-medical imaging. For some biological specimens, the thickness and refractive index inhomogeneity determine how much light scattering it produces. This class of specimens is referred to as phase objects, as they affect significantly only the phase of the input field rather than the amplitude. Too little scattering from the specimen makes it challenging to reveal the structure from an overwhelming input light background. If it is possible, it would be helpful if the authors can consider the reconfigurable image of pure-phase objects.

The quality of edge detection results can be described by resolution and contrast. For the proposed reconfigurable image processing metasurfaces. I suggest the authors add some quantitative results on the resolution and contrast of edge results. For edge-detected image [as shown in Figs.4 and 5], the imaging contrast dose not seem so good. The switching between edge detection and bright field imaging was achieved by temperature change. However, in the edge detection, the non-edge portion still maintains some visibility. Is it possible to improve the quality of edge detection by further suppressing the low-k component? The authors should explain the reason behind the phenomenon and how to improve it.

Reviewer #2 (Remarks to the Author):

The manuscript by Michele Cotrufo et al. have reported a reconfigurable image processing metasurfaces operating in the near infrared region whose response can be altered by temperature changes. The reconfigurability is achieved by utilizing the insulator-to-metal phase transition of a thin buried layer of vanadium dioxide, which in turn strongly alters the non-local response of the device. This work demonstrates the achievement of reconfigurable image processing metasurfaces with high numerical aperture, high efficiency, isotropy, and polarization independence. In general, the paper is scientific sound. However, there are still some content should be discussed clearly before I can recommend its publication in Nature Communications.

1. In Figure 1e, we can find that not both s- and p- polarizations the metasurface transfer function features a Laplacian profile when the VO₂ is in the insulating phase. Compared with the references [22] cited in this paper, the difference between the s-polarized and p-polarized transmission versus $kx = k_0 \sin \theta$ for $ky = 0$ is not neglectable. Please demonstrate whether the difference have an influence on the edge-enhanced image for unpolarized illumination.
2. In Supplementary Figure S1, the temperature of sample was increased by a thin ceramic theater with a thermal tape, which was heated via a temperature controller. Could the author provide the time of the switch between the insulator-to-metal phase transition and the repetition rate. Please provide the video of the image processing experimental results during the transition.
3. As an advantage of the image processing metasurface, polarization-independence may be better demonstrated if the authors could supplement the experimental result or numerical calculations of edge detection images for polarized illumination, which can be contrasted with the unpolarized one.
4. Please explain the limiting factors of the resolution of edge detection.
5. For whether the phase object edge can be detected, I hope the authors could discuss the possibility of such a scenario.
6. There are some suggestions for Figure 1, Figure 4 and Figure 5.
 - ①. Figure 1c shows calculated normal-incidence transmission spectrum of the metasurface. I think it's better to explain the polarization of the illumination in the explanation of Figure 1c
 - ②. Figure 4d and Figure 5c show the output intensity taken at the location marked by the white dashed line, however, the legend "T = 58°C(×6)" and "T = 53°C(×4)" are supposed to be explained in more detail.

Reviewer #3 (Remarks to the Author):

In this paper, the authors present a passive metasurface for edge detection operating in the near-infrared region, utilizing the characteristics of phase-change materials. The image processing effects of this metasurface can drastically change with small temperature variations, achieving numerous high optical metrics with potential applications in biomedical imaging and other fields. However, due to the high standards of Nature Communication, the authors need to address the following issues before my further consideration:

1. The authors assert that digital image processing suffers from high latency, energy consumption incompatible with independent devices, overall space occupancy, and complexity.

Therefore, adopting analog alternatives is attractive. I have the following questions regarding this work:

- Can the use of phase-change materials provide an advantage in terms of energy consumption? In this work, maintaining the metallic phase of the device requires temperatures above ~70 degrees. This poses a challenge not only in terms of energy consumption but also has negative implications for device heat dissipation.

- In signal processing, efforts should be made to minimize the impact of errors during processing. However, in the proposed solution, there inevitably exist unexpected processing outputs during the functional switching process. It would be better to discuss the specific switching time of functionalities to demonstrate the superiority of the proposed solution.

- For edge detection processing, the authors demonstrate the processing under different temperatures but do not show corresponding processing metrics such as edge processing Precision, Recall, Miss Rate, False Positive Rate, etc. It is challenging to illustrate the high level of edge processing.

- In Fig 4, showing the effect of edge processing with temperature variation, an uneven effect on image processing is observed in the temperature range of 50-60 degrees.

Could further analysis be conducted on this phenomenon?

- Further clarification is needed on the complexity of the device.

2. Reconfigurability is one of the core innovative points of the article, and therefore, further considerations are required.

- In the paper, the authors demonstrate the switching of two functionalities: edge enhancement and bright field. However, this is far from sufficient for reconfigurability. Can the authors at least demonstrate some possibilities to further extend the functionalities of this device?

3. I notice there are some tunable edge detection works (e.g., Nano Letters 21.20 (2021): 8715-8722). The authors should deeply compare them.

If the utilization of phase-change materials can introduce biggest inherent advantages.

Response to the Referees' Reports

Reviewer #1 (Remarks to the author):

Optical metasurfaces, as a class of optical metamaterials with a reduced dimensionality, have attracted much attentions due to their fascinating abilities of controlling light. Recently, metasurfaces have started making important progress in of analog computing and image processing. Various image processing and optical computing functionalities have been recently demonstrated, however, most of the considered devices are static and lack reconfigurability. In the present manuscript, the authors propose and demonstrate a reconfigurable image processing metasurface phase-change material. In the proposed scheme, the image processing response can be drastically modified by temperature variations around a CMOS-compatible temperature. In addition, this reconfigurability is accompanied by performance metrics close to optimal, and it is combined with a simple geometry compatible with large-scale manufacturing. There are several issues in the paper need to be elucidated further.

We thank the reviewer for their work on our manuscript and for providing useful feedback. In the following, we address all the reviewer's comments. For convenience, we have added numbering to each of the reviewer's comments.

- 1) In the section of Introduction, the authors write: Liquid crystals can also be used to achieve reconfigurable optical computation, although this approach typically requires thick devices used as spatially varying masks in 4f systems, thus limiting the miniaturization and integration opportunities. It should be noted that the fast switching with several millisecond switching time can be realized by liquid crystals. However, it is not clear how fast the switching time of the phase-change material is. If the switching time is not very fast, then what is the motivation of why a phase-change material is a good candidate in reconfigurable image?

We thank the reviewer for this question. Similar comments have also been raised by Reviewer 2 (comment #2) and Reviewer 3 (comment #1b).

For phase-change materials, the switching time is the sum of two components, $t_{switch} = t_{heat} + t_{PCM}$. The time t_{PCM} is the time taken for the phase-change material to transition from an insulator to a metal once the transition temperature has been achieved. This time is typically very short. Indeed, recent works [R1-R3] have found that the transition timescale of VO₂ is shorter than 1 ps, with record values of a few tens of femtoseconds reported in Ref. [R3]. Therefore, the intrinsic switching time of the phase-change material is short enough to perform fast (>GHz) signal processing. However, in practical devices the overall transition time is actually dictated by t_{heat} , which is the time required to change the temperature of the device from a temperature T_{cold} just below the transition temperature range to a temperature T_{hot} just above the transition range, or vice versa. In our experiment, the values of these temperatures were $T_{cold} \approx 60^\circ C$ and $T_{hot} \approx 68^\circ C$. The finite extent of this temperature range (as opposed to an abrupt transition temperature) is due to the inhomogeneities in the vanadium dioxide layer that induced small local variations of the transition temperature. Therefore, in our case, t_{heat} is the time required to change the metasurface temperature by approximately 8 degrees Celsius.

The speed at which the temperature change occurs depends on how the metasurface is heated. In our experiment, we heated the device with an external heater that was attached to the back-side of the device. Therefore, the timescale of the metasurface heating was dictated by the heat conduction through

the thick (1 mm) glass substrate, and hence it is expected to be fairly slow, likely on the ~ 1 second scale. However, it is important to emphasize that this is not an intrinsic limitation of the phase-change-material approach. More efficient heating mechanisms can be developed, for example by integrating a heating element directly on top of the metasurface. In Ref. [R4], for example, this approach was used to vary the temperature of a silicon metasurface by more than 50°C within a time shorter than 1 ms by using short 5 V voltage pulses. Moreover, phase-transition can also be induced electrically and without the need of any significant Joule heating effect [R5], which allows switching timescales of the order of hundreds of microseconds. Finally, all-optical heating, where a high-power laser alters the metasurface temperature, can be used to achieve even faster heating timescales. In Ref. [R6], for example, it was shown that a picosecond laser with an average power of $\sim 100\text{mW}$ can heat up a silicon metasurface by $\sim 40\text{K}$ within few tens of microseconds.

To address these points in the manuscript, we have added the following paragraph at page 14:

An important figure of merit for reconfigurable metasurfaces is the switching time, i.e. how long it takes for the metasurface to switch between two states once an external stimulus is applied. In our PCM-based device, the switching time is due to the sum of two components, $t_{\text{switch}} = t_{\text{heat}} + t_{\text{PCM}}$. The time t_{PCM} is the time that the phase-change material takes to transition from insulating to metal once the transition temperature is achieved. Several works [47], [48], [49] have found that t_{PCM} can be shorter than 1 ps, with values of the order of few tens of femtoseconds found in ref.[49]. These timescales would potentially allow using PCM-based metasurfaces to perform ultrafast signal processing. However, in practical devices the overall transition time is actually dictated by t_{heat} , which is the time required to change the sample temperature from a temperature T_{cold} to a temperature T_{hot} (or vice versa), where T_{cold} and T_{hot} are just below and above the transition range. In our experiment, $T_{\text{cold}} \approx 60^\circ\text{C}$ and $T_{\text{hot}} \approx 68^\circ\text{C}$, and thus t_{heat} is the time required to change the metasurface temperature by approximately 8 degrees Celsius. The speed at which this temperature change occurs depends on how the sample is heated. In our proof-of-principle experiment, we heat the sample with an external heater, which is in contact with the backside of the sample. Thus, the metasurface heating timescales are dictated by the heat conduction through the thick (1 mm) glass substrate, and therefore they are expected to be fairly slow, likely on the ~ 1 second scale. We note, however, that this is not an intrinsic limitation of this approach. More efficient heating mechanisms can be devised, for example by integrating metallic heating elements directly on top of the metasurface layer. In Ref. [50] this approach was used to vary the temperature of a silicon metasurface by more than 50°C within a time $< 1\text{ms}$ by using short 5V voltage pulses. Moreover, phase-transition can also be induced electrically and without the need of any significant Joule heating effect [51], leading to switching timescales of the order of hundreds of microseconds. Additionally, all-optical heating, where a high-power laser alters the metasurface temperature, can be used to achieve even faster heating timescales. In Ref. [52] it was shown that a picosecond laser with an average power of $\sim 100\text{mW}$ can heat up a silicon metasurface by $\sim 40\text{K}$ within few tens of microseconds.

Moreover, we have expanded some of the sentences in the Discussion:

Finally, in this device the temperature was controlled via external heater elements, which set hard bounds on the rate at which the temperature increases, and thus on the switching speed. However, the same working principle and metasurface design can be readily extended to devices where the metasurface temperature is controlled either by local heater elements integrated on the same chip [50][60], or via optically induced heating with an external pump laser [52].

[R1] Cavalleri, Andrea, et al. "Femtosecond structural dynamics in VO₂ during an ultrafast solid-solid phase transition." Physical review letters 87.23 (2001): 237401.

[R2] Wall, Simon, et al. "Tracking the evolution of electronic and structural properties of VO₂ during the ultrafast photoinduced insulator-metal transition." *Physical Review B* 87.11 (2013): 115126.

[R3] O'Callahan, Brian T., et al. "Inhomogeneity of the ultrafast insulator-to-metal transition dynamics of VO₂." *Nature communications* 6.1 (2015): 6849.

[R4] Zangeneh Kamali, Khosro, et al. "Electrically programmable solid-state metasurfaces via flash localised heating." *Light: Science & Applications* 12.1 (2023): 40.

[R5] Kabir, Sumaiya, et al. "Device Geometry Insights for Efficient Electrically Driven Insulator-to-Metal Transition in Vanadium Dioxide Thin-Films." *Advanced Electronic Materials* 8.1 (2022): 2100428.

[R6] Cotrufo, Michele, et al. "Passive bias-free non-reciprocal metasurfaces based on thermally nonlinear quasi-bound states in the continuum." *Nature Photonics* (2023): 1-10.

- 2) The authors described the fabrication of the imaging target, which is created by etching the desired shape on a 200 nm thick chromium layer deposited on a glass substrate. Therefore, it is known that the amplitude target has been applied in the reconfigurable imaging experiment. In the abstract, the authors write: Our work paves the way to a new generation of ultra-compact, tunable, passive devices for all-optical computation, with potential applications in augmented reality, remote sensing, and bio-medical imaging. For some biological specimens, the thickness and refractive index inhomogeneity determine how much light scattering it produces. This class of specimens is referred to as phase objects, as they affect significantly only the phase of the input field rather than the amplitude. Too little scattering from the specimen makes it challenging to reveal the structure from an overwhelming input light background. If it is possible, it would be helpful if the authors can consider the reconfigurable image of pure-phase objects.

We thank the reviewer for this suggestion (see also similar comment from Reviewer 2, comment #5). Indeed, the same metasurface that we used in our work could also be used for reconfigurable phase imaging of transparent samples that do not have any amplitude contrast. This stems from the fact that our metasurface performs the Laplacian operation on the amplitude of the input field (and not on its intensity). If the input field is given by $E_{in}(x, y) = A(x, y)e^{i\phi(x, y)}$, where $\phi(x, y)$ is the phase and $A(x, y)$ is the amplitude, then the output field (processed by the metasurface) is $E_{out}(x, y) = \nabla^2 E_{in}(x, y)$. For a pure phase sample (that is, if the amplitude is constant), $\nabla^2 A(x, y) = 0$ and hence $|E_{out}(x, y)|^2 = |A|^2 |\nabla^2 \phi(x, y)|^2$, i.e., the output field carries information on the phase gradients.

Phase imaging with meta-optical devices has been recently demonstrated in Refs. [R7] and [R8] with spatially non-local metasurfaces that have similar responses to that of our metasurface in the insulating state. Note that these references were cited in the manuscript (Refs. 20 and 23, respectively). To perform either edge detection or phase imaging, the metasurface needs to filter the spatial frequencies of the input image and suppress the low spatial frequencies. Therefore, our metasurface could be used for phase imaging in the insulating state and bright field imaging in the metallic state. Yet, the extension of these outcomes to phase imaging is beyond the scope of this manuscript.

To clarify the potential of our metasurface for phase imaging and to also address comment #5 of reviewer 2, we have added the following sentences:

- At the end of the introduction section, we added “*In particular, our device could also be used to perform reconfigurable phase contrast imaging [20], [23].*”
- At page 12, we added the paragraph:

We note that, while in our experiments we focused on edge detection, the same device can also be used to switch between phase contrast imaging [20], [23] and bright field imaging, which is of particular relevance for biological samples. This is possible thanks to the fact that, in these analog devices, the Laplacian operation is performed at the level of the field amplitude, i.e. $E_{out}(x, y) = \nabla^2 E_{in}(x, y)$. Thus, if the input field has a constant amplitude but a spatially varying phase, $E_{in}(x, y) = Ae^{i\phi(x,y)}$, the output field will be proportional to the Laplacian of $\phi(x, y)$, i.e. $E_{out}(x, y) \propto \nabla^2 \phi(x, y)$. Thus, the output intensity will carry information about the phase gradients and discontinuities of the input image.

[R7] Wesemann, L., Rickett, J., Song, J., Lou, J., Hinde, E., Davis, T.J. and Roberts, A., 2021. Nanophotonics enhanced coverslip for phase imaging in biology. *Light: Science & Applications*, 10(1), p.98.

[R8] Ji, A., Song, J.H., Li, Q., Xu, F., Tsai, C.T., Tiberio, R.C., Cui, B., Lalanne, P., Kik, P.G., Miller, D.A. and Brongersma, M.L., 2022. Quantitative phase contrast imaging with a nonlocal angle-selective metasurface. *Nature Communications*, 13(1), p.7848.

- 3) The quality of edge detection results can be described by resolution and contrast. For the proposed reconfigurable image processing metasurfaces. I suggest the authors add some quantitative results on the resolution and contrast of edge results. For edge-detected images [as shown in Figs.4 and 5], the imaging contrast does not seem so good. The switching between edge detection and bright field imaging was achieved by temperature change. However, in the edge detection, the non-edge portion still maintains some visibility. Is it possible to improve the quality of edge detection by further suppressing the low-k component? The authors should explain the reason behind the phenomenon and how to improve it.

We thank the reviewer for this suggestion. While we agree on the need to provide more quantitative information on resolution and contrast, we respectfully disagree with the reviewer that “the imaging contrast does not seem so good“. In Figs. 4 and 5, the contrast of the edges can be assessed quantitatively by looking at the horizontal slices of the output intensity provided in the accompanying 1D plots (see Fig. 4d and 5c). For convenience, we have reproduced Fig. 4d in this response letter (Fig. R1b below), together with numerical labels and a zoomed-in inset (Fig. R1a). From this figure it can be seen that the edge contrast, given by the ratio of the peak edge intensity and the intensity of the flat regions, is around 10. This value is similar to that obtained with other metasurfaces that lack reconfigurability, for example in Ref. 19: Y. Zhou, H. Zheng, I. I. Kravchenko, and J. Valentine, “Flat optics for image differentiation,” *Nat. Photonics*, vol. 14, no. 5, Art. no. 5, May 2020, doi: 10.1038/s41566-020-0591-3.

As the reviewer correctly points out, the unwanted background signal is determined by how well the metasurface can suppress low-k Fourier components. This is related to the level of normal-incidence

transmission of the metasurface in the insulating state. As shown in Fig. 3b, the transmission level for our device was about 2%-3%. Further reducing this transmission would in turn reduce the background and hence increase the contrast of the edges in the images.

Besides this background signal, some weaker oscillations are also visible in the experimental plots in Figs. 4 and 5 (in both input and output images), as also highlighted by the blue arrow and the blue shaded box in Fig. R1a. These peaks are not background or noise, and their origin is due to multiple effects. First, since we are imaging features with spatial scales ($\sim 10 \mu\text{m}$) close to the resolution limit set by the objective NA ($\text{NA}_{\text{obj}} = 0.42$, Resolution limit $\approx 2 \mu\text{m}$) and the metasurface NA ($\text{NA}_{\text{meta}} = 0.26$, Resolution limit $\approx 3.2 \mu\text{m}$, see also discussion below) we expect to observe some artifacts due to the clipping of the high-spatial-frequency components of the input image. These effects are particularly prominent here because our test input images contain sharp edges, that give rise to Fourier components with high spatial frequencies. We emphasize that this effect is not due to sub-idealities in the metasurface, as manifested by the fact that it is also visible in the input images (see Fig. R1c, which shows a zoomed-in view of Fig. 4a of the main paper, in the area where Fig. 4d was extracted). These artifacts become less prominent when images with large spatial sizes are processed.

The resolution of the edge-detected images is dictated by the largest spatial-frequency component that our metasurface can correctly process, i.e. by the NA of the metasurface. For our metasurface, a value of $\text{NA} = 0.26$ is estimated based on the angle-dependent transmission curves (see Figs. 1e and 3b). Following standard Fourier optics arguments, we expect the resolution limit of our metasurface to be $R_s = \frac{\lambda}{2\text{NA}} \approx 3.2 \mu\text{m}$. This value is consistent with the spatial features observed in the output images (see also Fig. R1), and with the fact that in the output images we can fully discern edges that are $\sim 10 \mu\text{m}$ apart from each other.

Fig. R1. Adapted from Fig. 4 of the main paper. The scale bar in panel (c) is equal to $7 \mu\text{m}$.

To address all of these comments in the manuscript, we implemented the following revisions:

- At page 13, we added the paragraphs

As shown in Fig. 4d the edge-detection contrast, defined as the ratio between the intensity of the detected edges and the intensity of the surrounding background areas, can be quite large, reaching values of about 10. The edge-detection contrast is directly related to how well the metasurface can suppress the low-spatial-frequency components of the input image, which in turn is dictated by the normal-incidence transmission level. In our device, we achieved normal-incidence transmissions of about 2%-3% (see Fig. 3b), mainly limited by the small absorption of the VO_2 in the insulating phase and by fabrication imperfections. Improvements of these factors would strongly reduce the residual background in the output

images. Figure 4d also reveals the presence of additional weaker intensity fluctuations between each pair of edges, both in the input and output images. Such weak intensity modulations are due to the fact that very-high spatial-frequency components, induced by the small size of the input image and the presence of sharp edges, are clipped by the finite objective NA (0.42) and metasurface NA (0.26). Nonetheless, the presence of these additional weaker oscillations does not limit the capability of performing edge detection, since the peaks corresponding to the edges are always much more intense.

- In the Discussion, we added the sentence:

The relatively large $NA \approx 0.26$ ensures that our metasurface can perform edge-detection with high spatial resolution, which can be estimated via $R \geq \frac{\lambda}{2NA} \approx 3.2 \mu\text{m}$. Indeed, in our processed images (see, e.g., Fig. 4c) we can fully resolve edges that are $\sim 10 \mu\text{m}$ apart from each other.

Reviewer #2 (Remarks to the Author):

The manuscript by Michele Cotrufo et al. have reported a reconfigurable image processing metasurfaces operating in the near infrared region whose response can be altered by temperature changes. The reconfigurability is achieved by utilizing the insulator-to-metal phase transition of a thin buried layer of vanadium dioxide, which in turn strongly alters the non-local response of the device. This work demonstrates the achievement of reconfigurable image processing metasurfaces with high numerical aperture, high efficiency, isotropy, and polarization independence. In general, the paper is scientific sound. However, there are still some content should be discussed clearly before I can recommend its publication in Nature Commuciations.

We thank the reviewer for their work on our manuscript, for providing constructive feedback, and for finding our work scientifically sound. In the following, we address all the reviewer's comments.

1. In Figure 1e, we can find that not both s- and p- polarizations the metasurface transfer function features a Laplacian profile when the VO₂ is in the insulating phase. Compared with the references [22] cited in this paper, the difference between the s-polarized and p-polarized transmission versus $kx = k_0 \sin\theta$ for $ky = 0$ is not neglectable. Please demonstrate whether the difference have an influence on the edge-enhanced image for unpolarized illumination.

We thank the reviewer for pointing out this potential source of misunderstanding. Indeed, Fig. 1e of our paper shows that the angle-dependent transmission of the metasurface follows the desired Laplacian-like behavior for both s- and p-polarized waves. However, the magnitude of the two curves is different, with p-polarized transmission being lower than the s-polarized transmission. For unpolarized light, which was also the case considered in our experiment, this effect does not have any impact on the uniformity and quality of the edge-enhanced images. This would remain true even in the extreme case where the p-polarized transmission remains zero over the entire angular range.

The role of the input polarization and the polarization-sensitivity of the metasurface was discussed in Ref. 24 (a paper authored by some of us). In Ref. 24, a metasurface with polarization asymmetry even larger than the one discussed in this paper was considered in its Figure 3, which is partially reproduced here in Fig. R2. In the device in Ref. 24, the p-polarized transmission (Fig. R2b, red line) remained very low, while

the s-polarized transmission (Fig. R2b, blue line) featured the desired angle-dependent profile. Nonetheless, when the input image was unpolarized (Fig. R2f), the edges of the input image were uniformly enhanced, independently of their orientation. This happens because the angular wave decomposition of any portion of an unpolarized input image contains an equal mixture of s- and p-polarized waves. The p-polarized contribution is uniformly reflected by the metasurface, while the s-polarized contribution undergoes the desired Laplacian filtering. This results in isotropic and direction-independent edge detection. This result remains true in the case (relevant to the device considered in our manuscript) in which the p-polarized transmission is non-zero but lower than the s-polarized transmission. In other words, an unpolarized image experiences an effective optical transfer function given by the average of the s- and p-polarized transfer functions.

Fig. R2. Adapted from Fig. 3 of ref. 24.

An asymmetry between the response to s- and p-polarized waves might have an impact in the case in which the input image is polarized. This was also shown in Fig. 3 of Ref. 24. In particular, some edges will be enhanced more than the others, depending on their orientation (see Figs. 3h and 3i of Ref. 24). However, these effects are expected to be small in our metasurface, since the difference in magnitude of the s- and p-polarized curves is not large. We demonstrate this explicitly in our answer to comment 3 of the same reviewer (see below), where the reviewer asked about the possible polarization-dependence of our device.

To clarify the issue raised in this comment, we have added the following paragraph at the end of the section “Design and Simulations”:

As shown in Fig. 1e, for both s and p impinging polarizations the metasurface transfer function features a flat profile when VO₂ is in the metallic phase. When the VO₂ is in the insulating phase, both s- and p-polarized transfer function feature the desired quasi-parabolic angle-dependent transmission, albeit with different magnitudes: the s-polarized transfer function reaches values larger than 0.8 at $k_x/k_0 = 0.25$, while the p-polarized transfer function is about 0.2 at the same excitation angle. As discussed in ref. [24], such polarization asymmetries do not introduce any practical issue in the quality and uniformity of the edge-detected image if the input image is

unpolarized. Instead, when the input image is linearly polarized, large asymmetries between s- and p-polarized transfer functions can lead to a scenario where different edges are enhanced with different efficiencies depending on their orientation [24]. In the experiments discussed here we focus on unpolarized input images. Nonetheless, in the Supplementary Information (Section S5) we show that, even when the input image is polarized, the polarization asymmetry of our device (Fig. 1e) is not large enough to introduce any visible inhomogeneity or anisotropy.

We have also included a new section in the Supplementary Information, as detailed in our answer to Comment #3 below.

2. In Supplementary Figure S1, the temperature of sample was increased by a thin ceramic theater with a thermal tape, which was heated via a temperature controller. Could the author provide the time of the switch between the insulator-to-metal phase transition and the repetition rate. Please provide the video of the image processing experimental results during the transition.

We thank the reviewer for raising this question. We note that reviewer 1 and reviewer 3 had similar comments.

Quantifying the switching time requires a careful analysis of the different timescales involved. In our metasurface, the switching time is the sum of two components, $t_{switch} = t_{heat} + t_{PCM}$. The time t_{PCM} is the time taken for the phase-change material to transition from an insulator to a metal once the transition temperature has been achieved. There have been works [R1-R3] that have found that this time is extremely short, i.e. shorter than 1 pico-second, with record values of a few tens of femtoseconds reported in Ref. [R3]. Therefore, the intrinsic switching time of the phase-change material is short enough to perform fast (>GHz) signal processing. However, the overall transition time in our device is actually dictated by t_{heat} , which is the time required to change the temperature of the device from a temperature T_{cold} just below the transition temperature to a temperature T_{hot} just above the transition, or vice versa. In our experiment, the values of these temperatures were $T_{cold} \approx 60^\circ C$ and $T_{hot} \approx 68^\circ C$. The finite extent of this temperature range, as opposed to an abrupt transition temperature, was due to the inhomogeneities in the vanadium dioxide layer that induced small local variations of the transition temperature. Therefore, the time t_{heat} in our case was the time required to change the temperature of the metasurface by approximately 8 degrees Celsius.

The speed that the temperature change occurs depends on how the metasurface is heated. In our experiment, we heated the device with an external heater that was attached to the back-side of the device. Therefore, the timescale of the metasurface heating was dictated by the heat conduction through the thick (1 mm) glass substrate and hence was relatively slower — on the order of about a second. However, this is not an intrinsic limitation of the phase-change material approach. More efficient heating mechanisms can be developed, such as integrating a heating element directly on the metasurface. This approach was used in Ref. [R4] to change the temperature of a silicon metasurface by more than $50^\circ C$ in less than a millisecond by applying short 5 V pulses. Moreover, phase-transition can be induced electrically and without the need of any significant Joule heating effect [R5], which allows switching timescales of the order of hundreds of microseconds. Another approach is to use a high-power laser to optically change the temperature of the metasurface. This was used for example in Ref. [R6] to achieve a faster rate of heating, where a pico-second laser with an average power of 100 mW heated a silicon metasurface by 40 K within tens of microseconds. Therefore, while our experiment did not focus on optimizing the overall switching time, it is possible to extend our method to devices with a faster change of temperature.

In our experiments, we repeated the heating and cooling process, and we observed that the process was repeatable. We have added a video of the image processing experiment for a single heating-cooling cycle in the Supplementary Information (Supplementary Video 1).

To address these points (and also comment #1 of reviewer 1 and comment #1b of reviewer 3), we have added the following paragraph at page 12 of the revised manuscript:

An important figure of merit for reconfigurable metasurfaces is the switching time, i.e. how long it takes for the metasurface to switch between two states once an external stimulus is applied. In our PCM-based device, the switching time is due to the sum of two components, $t_{\text{switch}} = t_{\text{heat}} + t_{\text{PCM}}$. The time t_{PCM} is the time that the phase-change material takes to transition from insulating to metal once the transition temperature is achieved. Several works [47], [48], [49] have found that t_{PCM} can be shorter than 1 ps, with values of the order of few tens of femtoseconds found in ref.[49]. These timescales would potentially allow using PCM-based metasurfaces to perform ultrafast signal processing. However, in practical devices the overall transition time is actually dictated by t_{heat} , which is the time required to change the sample temperature from a temperature T_{cold} to a temperature T_{hot} (or vice versa), where T_{cold} and T_{hot} are just below and above the transition range. In our experiment, $T_{\text{cold}} \approx 60^\circ\text{C}$ and $T_{\text{hot}} \approx 68^\circ\text{C}$, and thus t_{heat} is the time required to change the metasurface temperature by approximately 8 degrees Celsius. The speed at which this temperature change occurs depends on how the sample is heated. In our proof-of-principle experiment, we heat the sample with an external heater, which is in contact with the backside of the sample. Thus, the metasurface heating timescales are dictated by the heat conduction through the thick (1 mm) glass substrate, and therefore they are expected to be fairly slow, likely on the ~ 1 second scale. We note, however, that this is not an intrinsic limitation of this approach. More efficient heating mechanisms can be devised, for example by integrating metallic heating elements directly on top of the metasurface layer. In Ref. [50] this approach was used to vary the temperature of a silicon metasurface by more than 50°C within a time < 1 ms by using short 5V voltage pulses. Moreover, phase-transition can also be induced electrically and without the need of any significant Joule heating effect [51], leading to switching timescales of the order of hundreds of microseconds. Additionally, all-optical heating, where a high-power laser alters the metasurface temperature, can be used to achieve even faster heating timescales. In Ref. [52] it was shown that a picosecond laser with an average power of $\sim 100\text{mW}$ can heat up a silicon metasurface by $\sim 40\text{K}$ within few tens of microseconds.

Moreover, we have expanded some of the sentences in the Discussion:

Finally, in this device the temperature was controlled via external heater elements, which set hard bounds on the warming-up times, and thus on the switching speed. However, the same working principle and metasurface design can be readily extended to devices where the metasurface temperature is controlled either by local heater elements integrated on the same chip [50], [60], electrical effects [51], or via optically induced heating with an external pump laser [52].

[R1] Cavalleri, Andrea, et al. "Femtosecond structural dynamics in VO 2 during an ultrafast solid-solid phase transition." Physical review letters 87.23 (2001): 237401.

[R2] Wall, Simon, et al. "Tracking the evolution of electronic and structural properties of VO 2 during the ultrafast photoinduced insulator-metal transition." Physical Review B 87.11 (2013): 115126.

[R3] O'Callahan, Brian T., et al. "Inhomogeneity of the ultrafast insulator-to-metal transition dynamics of VO2." Nature communications 6.1 (2015): 6849.

[R4] Zangeneh Kamali, Khosro, et al. "Electrically programmable solid-state metasurfaces via flash localised heating." Light: Science & Applications 12.1 (2023): 40.

[R5] Kabir, Sumaiya, et al. "Device Geometry Insights for Efficient Electrically Driven Insulator-to-Metal Transition in Vanadium Dioxide Thin-Films." *Advanced Electronic Materials* 8.1 (2022): 2100428.

[R6] Cotrufo, Michele, et al. "Passive bias-free non-reciprocal metasurfaces based on thermally nonlinear quasi-bound states in the continuum." *Nature Photonics* (2023): 1-10.

3. As an advantage of the image processing metasurface, polarization-independence may be better demonstrated if the authors could supplement the experimental result or numerical calculations of edge detection images for polarized illumination, which can be contrasted with the unpolarized one.

We thank the reviewer for this suggestion (see also related comment #1 from the same reviewer). All our experiments were performed with unpolarized illumination. While we do not have measurements with polarized illumination, we can easily prove the polarization-independent behavior by performing numerical calculations as requested. In these calculations, we use the fully vectorial transfer functions from section S2 of the Supplementary Information to calculate the output image produced by the metasurface when input images with different states of linear polarization are used. The results of these calculations are shown in Fig. R3. We used an input image similar to the one considered in the experiments in Fig. 4. The calculations confirm that, for either unpolarized (Fig. R3b), x-polarized (Fig. R3c), or y-polarized (Fig. R3d) illumination, all the edges of the input images are enhanced uniformly and independently of their orientation. Moreover, the peak intensities of the three output images are almost identical. This confirms that the small polarization asymmetry observed in Fig. 1e does not introduce any sizeable polarization dependence in the image processing.

Fig. R3. (a) Input image. (b-d) Output image calculated with the transfer functions shown in Fig. S2, and assuming that the input image is (b) unpolarized, (c) x-polarized, (d) y-polarized.

In the revised paper, we have added a new section to the Supplementary Information (Section S4) to discuss the numerically calculated edge-detection results for different impinging polarizations, including Fig. R3. Moreover, we have added the following remark at the end of the “Design and Simulations” section:

In the experiments discussed here, we focus on unpolarized input images. For completeness, in the Supplementary Information (Section S5) we show that, even when the input image is polarized, the polarization asymmetry of our device (Fig. 1e) is not large enough to introduce any visible inhomogeneity or anisotropy in the output image.

4. Please explain the limiting factors of the resolution of edge detection.

The resolution of the edge-detected images is dictated by the largest spatial frequency component that our metasurface can correctly process, i.e. by the NA of the metasurface. For our device, a value of $NA = 0.26$ is estimated based on the angle-dependent transmission curves (see Figs. 1e and 3b). Following standard Fourier optics arguments, we expect the resolution limit of our metasurface to be $R_s = \frac{\lambda}{2NA} \approx 3.2 \mu\text{m}$. This value is consistent with the spatial features observed in the output images, and with the fact that in the output images, we can fully discern edges that are $\sim 10 \mu\text{m}$ apart from each other (see, e.g., Fig. 4c).

In the revised paper, we have added these sentences in the Discussion section

The relatively large $NA \approx 0.26$ ensures that our metasurface can perform edge-detection with high spatial resolution, which can be estimated via $R \geq \frac{\lambda}{2NA} \approx 3.2 \mu\text{m}$. Indeed, in our processed images (see, e.g., Fig. 4c) we can fully resolve edges that are $\sim 10 \mu\text{m}$ apart from each other.

5. For whether the phase object edge can be detected, I hope the authors could discuss the possibility of such a scenario.

We thank the reviewer for this suggestion (see also similar comment from Reviewer 1, comment #2). Indeed, the same device that we demonstrated in our work could also be used to perform reconfigurable phase imaging of samples that do not have any amplitude contrast. This can be intuitively understood from the fact that our computational metasurface applies the Laplacian operation on the amplitude of the input field, and not on its intensity. Assume that the input field is $E_{in}(x, y) = A(x, y)e^{i\phi(x, y)}$, where $\phi(x, y)$ is position-dependent phase factor and $A(x, y)$ is a real amplitude. Then, the output field is $E_{out}(x, y) = \nabla^2 E_{in}(x, y)$, thus, if the amplitude is constant (i.e., $\nabla^2 A(x, y) = 0$) the output field is $|E_{out}(x, y)|^2 = |A|^2 |\nabla^2 \phi(x, y)|^2$, i.e. the output field carries information on the phase gradients.

We note that phase imaging with meta-optical devices has been demonstrated recently [R7] [R8] with spatially non-local metasurfaces whose responses are very similar to that of our metasurface in the “cold” state. References R7 and R8 were already cited in the manuscript (references 20 and 23, respectively). In particular, to perform either edge detection or phase imaging the metasurface needs to filter the spatial frequencies of the input image, and selectively suppress low spatial frequencies. Thus, the metasurface investigated in this work would be able to perform phase imaging in the insulating state (room temperature) due to its ability to perform high-pass spatial frequency filtering. In its metallic state, on the other hand, the device would perform bright field imaging. We are actually working on an extension of this work to phase imaging, but this is outside of the scope of this manuscript.

In order to further clarify the potential of the devices for phase imaging (and also to address comment #2 of reviewer 1), we added the following sentences:

- At the end of the introduction section, we added “*In particular, our device could also be used to perform reconfigurable phase imaging [20], [23].*”
- On page 14, we added the paragraph:

We note that, while in our experiments we focused on edge detection, the same metasurface devices can also be used to switch between phase imaging [20], [23] and bright field imaging, which is of particular relevance for biological samples. This is possible thanks to the fact that, in these analog devices, the Laplacian operation is performed at the level of the field amplitude, i.e. $E_{out}(x, y) = \nabla^2 E_{in}(x, y)$. Thus, if the input field has a constant amplitude but a spatially varying phase, $E_{in}(x, y) = Ae^{i\phi(x, y)}$, the output field will be proportional to the Laplacian of $\phi(x, y)$, i.e. $E_{out}(x, y) \propto \nabla^2 \phi(x, y)$. Thus, the output intensity will carry information about the phase gradients and discontinuities of the input image.

[R7] Wesemann, L., Rickett, J., Song, J., Lou, J., Hinde, E., Davis, T.J. and Roberts, A., 2021. Nanophotonics enhanced coverslip for phase imaging in biology. *Light: Science & Applications*, 10(1), p.98.

[R8] Ji, A., Song, J.H., Li, Q., Xu, F., Tsai, C.T., Tiberio, R.C., Cui, B., Lalanne, P., Kik, P.G., Miller, D.A. and Brongersma, M.L., 2022. Quantitative phase contrast imaging with a nonlocal angle-selective metasurface. *Nature Communications*, 13(1), p.7848.

6. There are some suggestions for Figure 1, Figure 4 and Figure 5.

①. Figure 1c shows calculated normal-incidence transmission spectrum of the metasurface. I think it's better to explain the polarization of the illumination in the explanation of Figure 1c

We thank the reviewer for pointing out this potentially confusing point. We note that, since the simulation in Fig. 1c and the corresponding measurement in Fig. 2c were performed at normal incidence, and since the structure has a C6 rotational symmetry, **the transmission spectrum is polarization-independent**. That is, the normal-incidence transmission spectrum is the same for any incident polarization, and thus also for unpolarized light. In the simulations in Fig. 1c, we had fixed the impinging polarization to x, solely because in full-wave simulations any source needs to have a well-defined polarization. The same results are obtained for y-polarized excitation. In the measurements, we used unpolarized illumination.

To clarify these points, we implemented the following changes:

- In the caption of Fig. 1, describing panel c, we added the sentence:

These normal-incidence spectra are independent of the excitation polarization, due to the triangular rotational symmetry of the metasurface.

- In the text describing Fig. 1c, we added the following remarks:

“We note that, due to the C6 rotational symmetry of our design, the normal-incidence transmission spectra in Fig. 1c are independent of the polarization state of the excitation. However, polarization dependence is expected to arise as the polar angle θ is increased.”

- In the methods, when describing the measurement done to obtain Fig. 2c, we revised the relevant sentence in order to mention the light polarization,

“For the normal-incidence measurements shown in Fig. 2a, the output of an unpolarized and collimated broadband lamp was weakly focused on the metasurface, [...]”

②. Figure 4d and Figure 5c show the output intensity taken at the location marked by the white dashed line, however, the legend “T = 58 °C(×6)” and “T = 53 °C(×4)” are supposed to be explained in more detail.

We thank the reviewer for highlighting this potential source of confusion. The “(x6)” and “(x4)” labels mean that the intensities of the corresponding curves have been multiplied by a factor of x6 and x4, respectively. In the revised text, we have clarified the meaning of these labels in the caption of each figure.

Reviewer #3 (Remarks to the Author):

In this paper, the authors present a passive metasurface for edge detection operating in the near-infrared region, utilizing the characteristics of phase-change materials. The image processing effects of this metasurface can drastically change with small temperature variations, achieving numerous high optical metrics with potential applications in biomedical imaging and other fields. However, due to the high standards of Nature Communication, the authors need to address the following issues before my further consideration:

We thank the reviewer for their work on our manuscript and for providing useful feedback. In the following, we address all the reviewer’s comments.

1. The authors assert that digital image processing suffers from high latency, energy consumption incompatible with independent devices, overall space occupancy, and complexity. Therefore, adopting analog alternatives is attractive. I have the following questions regarding this work:

For convenience, we have added alphabetical labels to the following sub-comments of reviewer’s comment #1.

(A) Can the use of phase-change materials provide an advantage in terms of energy consumption? In this work, maintaining the metallic phase of the device requires temperatures above ~70 degrees. This poses a challenge not only in terms of energy consumption but also has negative implications for device heat dissipation.

We agree with the reviewer that, in our current implementation, a significant power consumption might be required to keep the device into the “bright field imaging” modality. The impact of this issue would highly depend on the specific application, and it can be mitigated by several factors. For example, one could envision applications where one of two image processing tasks (e.g., edge detection or bright field imaging) is only required for short amounts of time. In this scenario, the metasurface would be designed to perform such imaging task in the high-temperature state.

Moreover, our approach can be readily extended to the case of non-volatile phase-change materials [R10], such as Antimony Trisulfide (Sb_2S_3) [R9], $\text{Ge}_2\text{Sb}_2\text{Se}_4\text{Te}$ (GSST) [R10] and Sb_2Se_3 . These materials do not require a constant flow of heat to maintain a certain crystal phase. Instead, quick heating events,

controlled by short voltage pulses provided via integrated micro-heaters, can be used to trigger transitions between the two phases.

Thanks to the simplicity of our design, and to the fact only a thin unpatterned layer of PCM is required, it would be straightforward to replace the VO₂ used in our experiment with any other non-volatile PCMs.

To comment more explicitly on this possibility, we have revised a sentence in the Discussion,

Additionally, the approach demonstrated here can be extended to non-volatile PCMs [59], which would permit maintaining the desired functionality without actively heating the metasurface.

[R9] Delaney, Matthew, et al. "A new family of ultralow loss reversible phase-change materials for photonic integrated circuits: Sb₂S₃ and Sb₂Se₃." *Advanced Functional Materials* 30.36 (2020): 2002447.

[R10] Zhang, Yifei, et al. "Electrically reconfigurable non-volatile metasurface using low-loss optical phase-change material." *Nature Nanotechnology* 16.6 (2021): 661-666.

(B) In signal processing, efforts should be made to minimize the impact of errors during processing. However, in the proposed solution, there inevitably exist unexpected processing outputs during the functional switching process. It would be better to discuss the specific switching time of functionalities to demonstrate the superiority of the proposed solution.

We agree with the reviewer that the switching time is an important figure of merit for reconfigurable devices. Similar comments were raised also by reviewers 2 and 3.

Quantifying the switching time requires a careful analysis of the different timescales involved. In our metasurface, the switching time is the sum of two components, $t_{switch} = t_{heat} + t_{PCM}$. The time t_{PCM} is the time taken for the phase-change material to transition from an insulator to a metal once the transition temperature has been achieved. Previous works [R1-R3] have found that this time is typically shorter than 1 pico-second, with record values of a few tens of femtoseconds reported in Ref. [R3]. Therefore, the intrinsic switching times of the phase-change material are short enough to perform fast (>GHz) signal processing, and they are much shorter than other tuning mechanisms. The time t_{heat} is instead the time required to change the temperature of the device from a temperature T_{cold} just below the transition temperature to a temperature T_{hot} just above the transition, or vice versa.

In our proof-of-principle experiment, the overall transition time is dominated by t_{heat} , which is here defined as the time required to change the device temperature from $T_{cold} \approx 60^\circ C$ to $T_{hot} \approx 68^\circ C$. The speed that the temperature change occurs depends on how the metasurface is heated. In our experiment, we heated the device with an external heater that was attached to the back-side of the device. Therefore, the timescale of the metasurface heating was dictated by the heat conduction through the thick (1 mm) glass substrate and hence was relatively slower, on the order of about a second. However, this is not an intrinsic limitation of the phase-change material approach. More efficient heating mechanisms can be developed, such as integrating a heating element directly on the metasurface. This approach was used in Ref. [R4] to change the temperature of a silicon metasurface by more than 50 °C in less than a milli-second by applying short 5 V pulses. Another approach is to use a high-power laser to optically change the temperature of the metasurface. This was used for example in Ref. [R6] to achieve a faster rate of heating, where a pico-second laser with an average power of 100 mW heated a silicon metasurface by 40 K within tens of micro-seconds. Therefore, while our experiment did not focus on optimizing the overall switching time, it is possible to extend our method to devices with a faster change of temperature. Moreover, phase-transition can be

induced electrically and without the need of any significant Joule heating effect [R5], which allows switching timescales of the order of hundreds of microseconds.

We note that, even if this tuning approach requires a change of local temperature, the required temperature variations are much smaller than other temperature-based methods that rely solely on thermo-optic effects: In our device, a temperature change of $\sim 8^\circ\text{C}$ is enough to fully reconfigure the response. To obtain similar changes in the refractive index of silicon via thermo-optic effects, temperature variations larger than $\sim 100^\circ\text{C}$ would be required.

To address these points (and also similar comments from reviewers 1 and 2), we have added the following paragraph at page 12 of the revised manuscript:

An important figure of merit for reconfigurable metasurfaces is the switching time, i.e. how long it takes for the metasurface to switch between two states once an external stimulus is applied. In our PCM-based device, the switching time is due to the sum of two components, $t_{\text{switch}} = t_{\text{heat}} + t_{\text{PCM}}$. The time t_{PCM} is the time that the phase-change material takes to transition from insulating to metal once the transition temperature is achieved. Several works [47], [48], [49] have found that t_{PCM} can be shorter than 1 ps, with values of the order of few tens of femtoseconds found in ref.[49]. These timescales would potentially allow using PCM-based metasurfaces to perform ultrafast signal processing. However, in practical devices the overall transition time is actually dictated by t_{heat} , which is the time required to change the sample temperature from a temperature T_{cold} to a temperature T_{hot} (or vice versa), where T_{cold} and T_{hot} are just below and above the transition range. In our experiment, $T_{\text{cold}} \approx 60^\circ\text{C}$ and $T_{\text{hot}} \approx 68^\circ\text{C}$, and thus t_{heat} is the time required to change the metasurface temperature by approximately 8 degrees Celsius. The speed at which this temperature change occurs depends on how the sample is heated. In our proof-of-principle experiment, we heat the sample with an external heater, which is in contact with the backside of the sample. Thus, the metasurface heating timescales are dictated by the heat conduction through the thick (1 mm) glass substrate, and therefore they are expected to be fairly slow, likely on the ~ 1 second scale. We note, however, that this is not an intrinsic limitation of this approach. More efficient heating mechanisms can be devised, for example by integrating metallic heating elements directly on top of the metasurface layer. In Ref. [50] this approach was used to vary the temperature of a silicon metasurface by more than 50°C within a time < 1 ms by using short 5V voltage pulses. Moreover, phase-transition can also be induced electrically and without the need of any significant Joule heating effect [51], leading to switching timescales of the order of hundreds of microseconds. Additionally, all-optical heating, where a high-power laser alters the metasurface temperature, can be used to achieve even faster heating timescales. In Ref. [52] it was shown that a picosecond laser with an average power of $\sim 100\text{mW}$ can heat up a silicon metasurface by $\sim 40\text{K}$ within few tens of microseconds.

Moreover, we have expanded some of the sentences in the Discussion:

Finally, in this device the temperature was controlled via external heater elements, which set hard bounds on the warming-up times, and thus on the switching speed. However, the same working principle and metasurface design can be readily extended to devices where the metasurface temperature is controlled either by local heater elements integrated on the same chip [50], [60], electrical effects [51], or via optically induced heating with an external pump laser [52].

[R1] Cavalleri, Andrea, et al. "Femtosecond structural dynamics in VO 2 during an ultrafast solid-solid phase transition." Physical review letters 87.23 (2001): 237401.

[R2] Wall, Simon, et al. "Tracking the evolution of electronic and structural properties of VO 2 during the ultrafast photoinduced insulator-metal transition." Physical Review B 87.11 (2013): 115126.

[R3] O'Callahan, Brian T., et al. "Inhomogeneity of the ultrafast insulator-to-metal transition dynamics of VO₂." *Nature communications* 6.1 (2015): 6849.

[R4] Zangeneh Kamali, Khosro, et al. "Electrically programmable solid-state metasurfaces via flash localised heating." *Light: Science & Applications* 12.1 (2023): 40.

[R5] Kabir, Sumaiya, et al. "Device Geometry Insights for Efficient Electrically Driven Insulator-to-Metal Transition in Vanadium Dioxide Thin-Films." *Advanced Electronic Materials* 8.1 (2022): 2100428.

[R6] Cotrufo, Michele, et al. "Passive bias-free non-reciprocal metasurfaces based on thermally nonlinear quasi-bound states in the continuum." *Nature Photonics* (2023): 1-10.

(C) For edge detection processing, the authors demonstrate the processing under different temperatures but do not show corresponding processing metrics such as edge processing Precision, Recall, Miss Rate, False Positive Rate, etc. It is challenging to illustrate the high level of edge processing.

While we agree with the reviewer that several metrics would have to be assessed when using these devices in specific applications, we argue that such analysis is outside of the scope of this manuscript. Our work demonstrates analog and reconfigurable computation without any electronic step and at the speed of light. Here, edge detection is obtained as the outcome of a specific image processing operation. The limitations and relevant metrics of this process will be similar to those seen in the digital equivalent, with the addition of artifacts associated with deviations from the optimal transfer function.

Ultimately, performance metrics will depend on the types of images investigated, the specific transfer function implemented, and the overall optical system (including light sources and the camera) used for image formation. We note that some of the metrics mentioned by the reviewer are connected to the efficiency metric that we discuss in our manuscript: for example, a device with a high-intensity efficiency is capable of detecting edges characterized by weak peak intensities and weak intensity gradients.

Our work experimentally demonstrates the potential for using metasurfaces to undertake edge detection, and we note that other mathematical operations for performing this and other types of image analysis can also be implemented using metasurfaces.

(D) In Fig 4, showing the effect of edge processing with temperature variation, an uneven effect on image processing is observed in the temperature range of 50-60 degrees. Could further analysis be conducted on this phenomenon?

We assume that the reviewer is referring to the fact that, for certain temperatures (e.g. $T = 64\text{ }^{\circ}\text{C}$ in Fig. 4e and $T = 55\text{ }^{\circ}\text{C}$ in Fig. 4f) the output image is a bright-field image but with regions of uneven intensity. This effect can be attributed to the some inhomogeneities in the VO₂ layer during the sputter deposition process, which induce local variations of the transition temperature. As a result, different portions of the metasurface will experience the insulating-to-metal transition at slightly different temperatures, resulting in some image distortion when the temperature lies in the transition range [$61\text{ }^{\circ}\text{C} - 68\text{ }^{\circ}\text{C}$]. These effects can be mitigated by optimizing surface characteristics of VO₂ thin films.

To comment on this, we added the following sentences on page 13:

For temperatures within the transition region some image distortion is noticeable in the output images (e.g. $T = 64\text{ }^{\circ}\text{C}$ in Fig. 4e and $T = 55\text{ }^{\circ}\text{C}$ in Fig. 4e). This effect can be attributed to the aforementioned inhomogeneities in the VO_2 film that introduce local variations of the threshold temperature [43]. As a result, different portions of the metasurface will experience the insulating-to-metal transition at slightly different temperatures, resulting in an undesired image distortion.

(E) Further clarification is needed on the complexity of the device.

We are not sure what the reviewers mean with “complexity of the device”. Our proposed device is based on a very simple design and working principle. For this specific demonstration, we utilized a single-layer silicon metasurface, composed of a triangular array of etched holes. However, our approach is agnostic to the specific metasurface design, and it could be readily extended to other designs, such as the ones in Ref. 17 or 19. To make the metasurface response reconfigurable, we added a thin layer of VO_2 underneath the metasurface. Reconfigurability is then achieved by varying the device temperature and by leveraging the phase transition of the VO_2 . In our experiment, the temperature was changed with an external heater, but the device could also be equipped with on-chip micro-heaters to speed up the temperature change, as discussed in our answer to comment 1b.

Moreover, as we mention also in the answer to the next comment, our design does not require etching the VO_2 and it relies on very thin films. This makes our approach easily extendable to any other phase-change material. Thus, the design and working principle of our device could be immediately extended to different spectral regions and material platforms. Indeed, we argue that its extreme simplicity and broad applicability are the main advantages of our device.

To further clarify this, we have expanded some of the sentences in the Discussion:

Finally, in this device the temperature was controlled via external heater elements, which set hard bounds on the warming-up times, and thus on the switching speed. However, the same working principle and metasurface design can be readily extended to devices where the metasurface temperature is controlled either by local heater elements integrated on the same chip [50], [60], electrical effects [51], or via optically induced heating with an external pump laser [52].

2. Reconfigurability is one of the core innovative points of the article, and therefore, further considerations are required.

- In the paper, the authors demonstrate the switching of two functionalities: edge enhancement and bright field. However, this is far from sufficient for reconfigurability. Can the authors at least demonstrate some possibilities to further extend the functionalities of this device?

We thank the reviewer for this inspiring comment. Our device allows switching and reconfiguring the response of a computational metasurface between two very distinct states. While we agree with the reviewer that our proof-of-concept device does not allow arbitrary reconfigurability, we nonetheless believe that it represents an important step forward towards more complex functionalities, especially because of the simplicity of the design and the broad applicability of the working principle.

In the current implementation, we have demonstrated a metasurface that allows switching between edge enhancement and bright field imaging. In other words, in the high-temperature state, the metasurface applies the identity operation on the input image. More complex functionalities can be obtained by designing the metasurface such that, when the phase-change material has transitioned into a different phase, the transfer

function features a strongly angular-dependent response (instead of the almost-flat response in our Figs. 1(c-e)), in order to encode an image processing task different from the identity operator.

In this experiment, we have used a phase-change material (VO_2) which features an insulating-to-metallic phase transition. In the metallic state, the large loss of VO_2 prevents the build-up of nonlocal effects in the metasurface, which are essential to obtain nontrivial angle-dependent responses. However, due to the simplicity and universality of our approach, other phase-change materials – with lower loss – could be readily embedded within our metasurface. For example, Antimony Trisulfide (Sb_2S_3) [R10] features an amorphous-to-crystalline phase transition characterized by a large change of the real part of the refractive index $\Delta n = n_{\text{crystalline}}/n_{\text{amorphous}} > 1.2$ at $\lambda \approx 1550$ nm, while the imaginary part of the refractive index remains $\kappa < 10^{-5}$ in both states. Such low loss would allow designing of a metasurface with strong nonlocal effects for both phases of Sb_2S_3 . As a result, a nontrivial image processing task would be achieved in both phases.

To comment more on this, we have added the following sentences to the Discussion section,

Further improvement of the proposed design could lead to the implementation of more complex responses whereby, for example, the metasurface performs two different mathematical operations of choice in the two states. To obtain a more interesting functionality at high temperatures (as opposed to the bright-field imaging demonstrated here), the metasurface must support a nontrivial angle-dependent transfer function also when the PCM has transitioned to the high-temperature state. In the current experiment, the angle-dependent transfer function at high temperatures (Figs. 1e and 3b) is limited by the large loss induced by the metallic phase of the VO_2 . PCMs with different types of phase transitions could be used instead of VO_2 in order to minimize the impact of loss. For example, Antimony Trisulfide (Sb_2S_3) features an amorphous-to-crystalline phase transition characterized by a large change of the real part of the refractive index ($\Delta n = n_{\text{crystalline}}/n_{\text{amorphous}} > 1.2$ at $\lambda \approx 1550$ nm), while the imaginary part of the refractive index remains $\kappa < 10^{-5}$ in both states [58].

[R10] Delaney, Matthew, et al. "A new family of ultralow loss reversible phase-change materials for photonic integrated circuits: Sb_2S_3 and Sb_2Se_3 ." *Advanced Functional Materials* 30.36 (2020): 2002447.

3. I notice there are some tunable edge detection works (e.g., *Nano Letters* 21.20 (2021): 8715-8722). The authors should deeply compare them. If the utilization of phase-change materials can introduce biggest inherent advantages.

We would like to emphasize that the paper mentioned by the reviewer was already cited in our manuscript (Ref. 26 in the original manuscript) and we had already briefly commented on the comparison between that work (and other works) with ours. The text in the original version of the manuscript read

For example, Zhang et al. [26] have demonstrated computational metasurfaces reconfigurable through mechanical strain, yet with limitations in terms of reproducibility and scalability of such an approach to reconfigurable image processing. Several theoretical proposals [27], [28], [29], [30] have suggested that large reconfigurability may be achieved by electrically gating a graphene layer placed above or inside a metasurface. Liquid crystals can also be used to achieve reconfigurable optical computation, although this approach typically requires thick devices used as spatially varying masks in 4f systems [31], [32], thus limiting the miniaturization and integration opportunities. Material nonlinearities have also been proposed as a tool to achieve reconfigurability, either by having a high-intensity image impinging on a metasurface made of a

saturable absorber [33], or by controlling the metasurface response via an external optical pump [34].

While utilizing mechanical strain is an interesting and creative approach, we argue that this has limitations for practical applications. Indeed, if one wants to use the approach in Ref. 26 in integrated systems, it would be necessary to induce a controllable and accurate mechanical movement on the micro-scale. In the experiment in Ref. 26, this was achieved with macroscopic translational stages. While the required mechanical strain could be induced with integrated micro-electro-mechanical systems, it would also greatly increase the complexity of the device and therefore hinder its miniaturization and scalability. On the contrary, our approach does not require any complicated electronic circuits and mechanical components, apart from the optional use of metallic heaters to speed up the switching time.

In the revised manuscript, besides the paragraph mentioned above, we have also added a remark in the Discussions,

Our approach and design make this reconfigurable image-processing metasurface amenable to mass manufacturing. In particular, different from previous proposals, our approach does not require any mechanical and/or moving parts, electrical biases, or high-power optical excitations.

REVIEWERS' COMMENTS

Reviewer #1 (Remarks to the Author):

In the revised version of the manuscript, the authors have done a commendable job at shedding lights upon the questions raised during the first review process. This not only makes the manuscript more insightful, but also makes the whole procedure easier to follow. As have been mentioned in the previous review, the authors have proposed and demonstrated a reconfigurable image processing metasurface with phase-change material. In the proposed scheme, the image processing response can be drastically modified by temperature variations around a CMOS-compatible temperature. The results of this study are well demonstrated and could have potential benefits to the research on metasurfaces for reconfigurable all-optical computing and image processing. Therefore, the paper can now be accepted for publication.

Reviewer #2 (Remarks to the Author):

The authors have answered my questions properly.

Reviewer #3 (Remarks to the Author):

I have meticulously examined the authors' responses to all reviewers and thoroughly re-evaluated the manuscript. The revisions made by the authors effectively addressed my concerns and implemented necessary changes to enhance the quality of the paper. I am delighted to recommend the publication of this manuscript.